# Learning Ante-hoc Explanations for Molecular Graphs

## Abstract

Explaining the decisions made by machine learning models for high-stakes applications is critical for transparency. This is particularly true in the case of models for graphs, where decisions depend on complex patterns combining structural and attribute data. We propose EAGER (Effective Ante-hoc Graph Explainer), a novel and flexible ante-hoc explainer designed to discover explanations for graph neural networks, with a focus on the chemical domain. As an ante-hoc model, EAGER inductively learn a graph predictive model and the associating explainer together. We employ a novel bilevel iterative training process based on optimizing the Information Bottleneck principle, effectively distilling the most useful substructures while discarding irrelevant details. As a result, EAGER can identify molecular substructures that contain the necessary and precise information needed for prediction. Our experiments on various molecular classification tasks show that EAGER explanations are better than existing post-hoc and ante-hoc approaches.

## 1 Introduction

A critical problem in machine learning on graphs is understanding predictions made by graph-based models in high-stakes applications. This has motivated the study of graph explainers, which aim to identify subgraphs that are both compact and correlated with model decisions. However, there is no consensus on what constitutes a good explanation. Recent papers (Ying et al., 2019; Luo et al., 2020; Bajaj et al., 2021) have proposed different alternative notions of explainability that do not consider the user and instead are validated using examples. Other approaches have applied labeled explanations to learn an explainer directly from data (Faber et al., 2020). However, datasets with such labeled explanations are usually not available.

Explainers can be divided into *post-hoc* and *ante-hoc* (or intrinsic) (Vilone & Longo, 2020). Post-hoc explainers treat the prediction model as a black box and learn explanations by modifying the input of a pre-trained model (Yuan et al., 2022). On the other hand, ante-hoc explainers learn explanations as part of the model. Figure 2 compares the post-hoc and ante-hoc approaches in the context of graph classification. Even though post-hoc explainers are flexible in that they make no assumption about the model to be explained, there are two major limitations: (1) they are not sufficiently informative to enable the user to reproduce the behavior of the model, and (2) they are often based on a model that was trained without taking explainability into account. Ante-hoc explainers do not suffer from these limitations. Because the explanations are inductively learned together with the model, they are part of the prediction and hence reproducible.

A number of ante-hoc explainers on graphs have been proposed for graph learning (Miao et al., 2022; Yu et al., 2020; Chen et al.; Sui et al., 2022). CAL Sui et al. (2022) learns to recognize graphs with causally attended patterns and trivially attended patterns. However, this method relies on the assumption that causal and shortcut features exist. Additionally, modeling these features adds to the learning complexity. Other methods, like GSAT (Miao et al., 2022) and GIB (Yu et al., 2020) apply the Information Bottleneck (IB) principle (Tishby et al., 2000) to discover explanatory subgraphs. IB finds a balance between compressing information and retaining relevant information for a specific task by distilling the essential features from a dataset that are most relevant to predicting a target variable, while discarding unnecessary details. In this way, the compressed representation explains the information the model uses to arrive at a prediction. Unfortunately, directly estimating and optimizing the IB objective is difficult in the graph space due to the intractability of computing

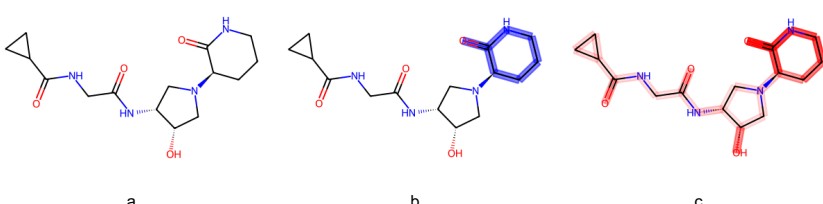

a        b        c

Figure 1: An example of explanations generated by our approach (EAGER) on molecular graphs from the synthetic dataset LACTAM. Subfigure (a) shows an input molecule, subfigure (b) highlights the ground-truth explanation and subfigure (c) illustrates the edge weights assigned by EAGER. Our method correctly identified the subgraph corresponding to the ground-truth lactam functional group.

the mutual information between graph distributions. Existing methods (Miao et al., 2022; Yu et al., 2020) overcome this problem by adopting variational bounds to optimize the IB objectives. However, defining an appropriate variational distribution is difficult in the graph space and one has to make simplifying assumptions regarding features and edge independence. This can lead to a loose approximation due to the complexity of the graph space. We take a different approach, optimizing the IB objectives via iterative bilevel training. Our method is theoretically inspired by the seminal work of Tishby et al. (2000), which proposes a system of self-consistent equations resulting from the IB objectives and an iterative process that has been shown to yield a local optima (Tishby & Zaslavsky, 2015).

We introduces EAGER (Effective Ante-hoc Graph Explainer), a novel ante-hoc explainer for graphs inspired by the Information Bottleneck (IB) principle. EAGER finds compact explanations while maximizing the graph classification accuracy. We formulate the training as a bilevel optimization problem. The inner loop optimizes the predictions given the explanations while the outer loop selects the most relevant explanation. Because our explanations are given as input to the GNN, no information outside of the explanation is used for prediction. Moreover, our explanations are learned jointly with the GNN, which enables EAGER to learn GNNs that are accurate and explainable. In fact, we show that EAGER's explainability objective produces an inductive bias that often improves the accuracy of the learned GNN compared to the base model. We emphasize that while EAGER is an ante-hoc model, it is general enough to be applied to a broad class of GNNs.

Figure 1 shows an example of EAGER explanations on an input molecule from LACTAM, one of our semi-synthetic molecular datasets. The goal is to identify whether the molecule contains lactam functional groups, which are cyclic amides of various ring sizes. Figure 1b shows the molecule with the ground-truth explanation highlighted in blue. Figure 1c illustrates the edge influences learned by EAGER, with darker shades of red indicating higher weights or more important edges. As expected, edges corresponding to the ground-truth explanation receive distinguishably higher weights. We show more examples of explanations by EAGER in Appendix G. To summarize:

- We propose EAGER, a novel framework for learning explainable GNNs. EAGER applies bilevel optimization to solve the IB problem, maximizing the mutual information between the explanation and the prediction in the inner loop while minimizing that between the explanation and the input graph, effectively eliminating irrelevant information. This adaptation of IB is different from existing literature that typically resorts to variational bounds.

- We compare EAGER against state-of-the-art graph classification and GNN explainer baselines using 8 datasets—including 5 real-world ones. EAGER not only performs competitively or better compared to the baselines in terms of accuracy in most settings but also generates precise and reproducible explanations. We further provide insights on the mutual information between input graphs and explanations, an analysis neglected by previous works.

- We propose three new semi-synthetic datasets with ground-truth explanations. These datasets are composed from real-world molecules.

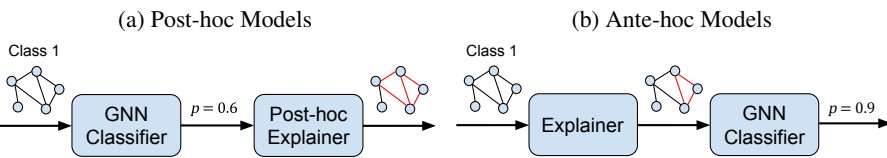

Figure 2: (a) Post-hoc models generate explanations for a pre-trained GNN classifier using its predictions. (b) Ante-hoc models, as our approach, learn GNNs and explanations jointly. This enables ante-hoc models to identify GNNs that are both explainable and accurate.

## 2 RELATED WORKS

**Explainability of GNNs:** Explainability has become a key requirement for the application of machine learning in many settings (e.g., healthcare, court decisions, scientific discoveries) (Molnar, 2020). Several post-hoc explainers have been proposed for explaining GNNs' predictions using subgraphs (Ying et al., 2019; Luo et al., 2020; Yuan et al., 2021; Bajaj et al., 2021; Tan et al., 2022; Xie et al., 2022). GNNExplainer (Ying et al., 2019) applies a mean-field approximation to identify subgraphs that maximize the mutual information with GNN predictions. PGExplainer (Luo et al., 2020) applies a similar objective, but samples subgraphs using the *reparametrization trick*. RCExplainer (Bajaj et al., 2021) generates counterfactual subgraph explanations. While post-hoc explainers treat a trained GNN as a black box —i.e., it only relies on predictions made by the GNN—ante-hoc explainers are model-dependent. DIR-GNN trains an intrinsically explainable GNN by discovering and generating label-invariant rationales (subgraphs). GIB (Yu et al., 2021) and GSAT (Miao et al., 2022) apply the information Bottleneck principle to learn subgraphs relevant for classification. MixupExplainer Zhang et al. (2023) enhances GIB's concept with mixup techniques to produce explainable graph augmentations that further improve predictive performances. Interestingly, some other works turn to the concept of causality for explanation Sui et al. (2022); Lin et al. (2022). Among them, OrphicX Lin et al. (2022) is a post-hoc method in which causal features features are used to generatively construct causal explanatory graphs. CAL Sui et al. (2022) is an ante-hoc model that learns to distinguish causally attended graphs and trivially attended graphs.

**Information Bottleneck principle:** Information Bottleneck (IB) principle finds a balance between compressing information and retaining relevant information for a specific task by distilling the essential features from a dataset that are most relevant to predicting a target variable, while discarding unnecessary details. This principle is has been widely used in machine learning, especially in deep learning, to improve model efficiency and generalization (Amjad & Geiger, 2019; Goldfeld & Polyanskiy, 2020; Kawaguchi et al., 2023; Saxe et al., 2019; Tishby & Zaslavsky, 2015). The IB principle has gained significant attention in the interpretation of GNNs. Miao et al. (2022) proposed GSAT, which uses IB to construct ante-hoc explainers. GSAT controls the mutual information (MI) between input graphs $G$ and explanatory subgraphs $S$ via a stochastic attention mechanism. Instead of minimizing $I(S; G)$, a variational upper bound is minimized. The family of distributions for the variational upper bound is constructed using a model in which edges are added between two nodes based on a Bernoulli distribution. IB-subgraph (Yu et al., 2020) also generates explanations using IB by adopting a bilevel optimization where $I(S, G)$ is estimated in the inner loop and maximizing the mutual information between $S$ and the classification outcome Y in the outer loop. Due to the difficulty of estimating $I(S, G)$, the method utilizes sampling to compute an upper bound. Chen et al. adopts a black box approach to explaining GNNs and focus on the Out of Distribution problem.

**Bilevel optimization:** Bilevel optimization is a class of optimization problems where two objective functions are nested within each other (Colson et al., 2007). Although the problem is known to be NP-hard, recent algorithms have enabled the solution of large-scale problems in machine learning, such as automatic hyperparameter optimization and meta-learning (Franceschi et al., 2018). Bilevel optimization has also been applied to graph problems, including graph meta-learning (Huang & Zitnik, 2020) and transductive graph sparsification (Wan & Schweitzer, 2021).

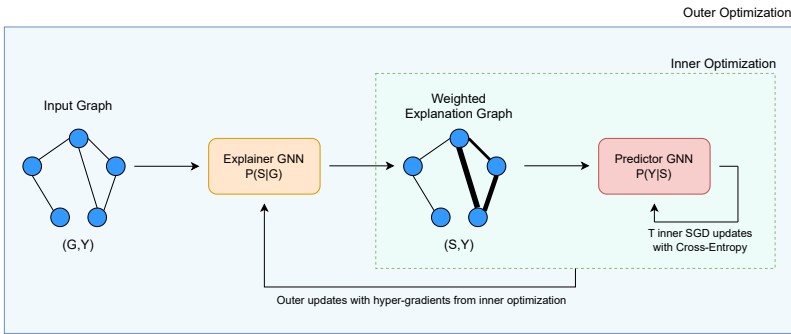

Figure 3: Illustration of the EAGER framework. The GNN explainer generates an explanation graph from the input graph by assigning an influence value to each edge. Edge influences are incorporated as edge weights on the explanation graph, the input of the predictor GNN. The inner problem optimizes the predictor GNN with $T$ iterations, while the outer problem updates the GNN explainer using hypergradients from inner iterations. The subgraph corresponding to the highly weighted edges dominate the final explanation graph embeddings, which is used in label prediction.

## 3 METHOD

### 3.1 PROBLEM FORMULATION

We formulate our problem as a supervised graph classification. A graph $G = (V, E)$ has node attributes $x_v$ for $v \in V$ and edge attributes $e_{uv}$ for $(u, v) \in E$. Given a graph set $\mathcal{G} = \{G_1, G_2, \ldots, G_n\}$ and continuous or discrete labels $\mathcal{Y} = \{y_1, y_2, \ldots, y_n\}$ for each graph respectively, our goal is to learn a function $\hat{f} : \mathcal{G} \to \mathcal{Y}$ that approximates the labels of unseen graphs.

### 3.2 INFORMATION BOTTLENECK PRINCIPLE

The Information Bottleneck (IB) Principle (Tishby et al., 2000) offers a compelling framework for understanding how to extract relevant information from data while compressing unnecessary details. This principle emphasizes the balance between preserving essential information and reducing complexity. The main idea is to find the most relevant information about a target variable from an input variable, while minimizing the amount of information retained about the input itself. In the case of graph learning, this idea is equivalent to finding the most predictive subgraphs by removing irrelevant structures.

Formally, given an input graphs $G$ and the target labels $Y$, the goal of the IB framework is to find a subgraph $S$ of $G$ that maximizes the mutual information $I(S; Y)$ while minimizing the mutual information $I(S; G)$. This trade-off can be expressed as an minimization problem, balancing the retention of relevant information about $Y$ against the compactness of $S$:

$$S^* = \arg\min_S -I(S; Y) + \beta I(S; G), \tag{1}$$

where $\beta$ is a Lagrange multiplier that controls the trade-off between the two information terms. To solve the above optimization, Tishby et al. (2000) proposed an iterative algorithm that aim to refine the representation of the input data at every step:

$$\begin{cases} p_t(S_i|G_i) = \frac{p_t(S_i)}{M_t} \exp^{-\beta D_{KL}[P(Y_i|S_i)||P(Y_i|G_i)]} \\ p_{t+1}(S_i) = \sum_{G_i} p(G_i) p_t(S_i|G_i) \\ p_{t+1}(Y_i|S_i) = \sum_{G_i} p(Y_i|G_i) p_t(G_i|S_i), \end{cases} \tag{2}$$

in which $M_t$ is a normalization term at each iteration. Tishby & Zaslavsky (2015) showed that a locally optimal solution can be found by iteratively solving a set of self-consistent equations, resulting in the above iterative scheme to solve the IB objective. At each step, two distributions $p(S)$ and $p(Y|S)$ are calculated and kept constant while the algorithm finds the optimal third distribution

$p(S|G)$ via a closed-form formula. This formula is obtained from taking the derivative of the IB-objective in Equation 1 with respect to $p(S|G)$. Since the IB-functional is not convex in the product space of these three sets, different initialization might lead to different local optima.

There are challenges to adopting the iterative IB method for graphs, since the input is no longer i.i.d. and because of the aforementioned initialization problem. We adopt bilevel optimization for the iteration shown in equations 2. The inner loop tunes the weights of the GNN so that its representation is consistent with the current explainer and uses it to compute the KL-divergence between $p(Y|S)$ and $p(Y|G)$. This step leads to an update of $p(S|G)$ and $p(S)$ in the outer loop, resulting in updated explanations. The process iterates until convergence.

We would like to further contrast our approach with existing works. In equation 1, the optimization of the first term can be done by computing the cross-entropy loss with respect to the labels. It is the second term, $I(S; G)$, that presents the main difficulty. Existing methods such as GSAT use a variational bound by defining a family of surrogate distributions that are generated in the input space of graphs. Our approach is different. We compute an approximation in the representation space of the graph data by computing the KL-divergence between $p(Y|S)$ and $p(Y|G)$, and using it to update $p(S)$ iteratively. Our novelty is in developing another alternative to approximating $I(S; G)$, and as we show later, for the datasets we consider, it achieves better results than those of the current variational approaches.

### 3.3 EAGER: EFFECTIVE ANTE-HOC GRAPH EXPLAINER

We introduce EAGER, an ante-hoc explainer that generates explanations by optimizing the IB principle via a bilevel process. EAGER performs compact and discriminative subgraph learning as part of the GNN training that optimizes the prediction of class labels in graph classification tasks. Our framework is illustrated in Figure 3. The explainer assigns an influence value to each edge, which will be incorporated into the original graph. The GNN classifier is trained with this new graph over $T$ inner iterations. Gradients from inner iterations are kept to update the explainer in the outer loop. The outer iterations minimize a loss function that induces the explanations to be discriminative. We will now describe our approach in more detail.

#### 3.3.1 EXPLAINER: SUBGRAPH LEARNING

EAGER is an edge-based subgraph learner. It learns edge representations from edge features and the corresponding node representations/features. Surprisingly, most edge-based explainers for undirected graphs are not permutation invariant when calculating edge representations (e.g., PGExplainer (Luo et al., 2020) concatenates node representations based on their index order). Shuffling nodes could change their performance drastically since the edge representations would differ. We calculate permutation invariant edge representations $h_{ij}$ given two node representations $h_i$ and $h_j$ as follows: $h_{ij} = [\mathbf{max}(h_i, h_j); \mathbf{min}(h_i, h_j)]$, where $\mathbf{max}$ and $\mathbf{min}$ are pairwise for each dimension and $[\cdot; \cdot]$ is the concatenation operator. Additionally, most existing explainers do not consider edge features, which are crucial for certain domains such as molecular graphs. Since many recent GNNs readily incorporate edge features (Hu et al., 2020b), we utilize them in processing the input graph to obtain $h_i$ and $h_j$ that take into account both node and edge features. Edge influences are learned via an MLP with sigmoid activation based on edge representations: $z_{ij} = MLP(h_{ij})$. This generates an edge influence matrix $Z \in [0, 1]^{n \times n}$. We denote our explainer function as $g_\Phi$ with trainable parameters $\Phi$.

#### 3.3.2 PREDICTOR: INFLUENCE-WEIGHTED GRAPH NEURAL NETWORKS

Any GNN architecture can be made sensitive to edge influences $Z$ via a transformation of the adjacency matrix of the input graphs. As our model does not rely on a specific architecture, we will refer to it generically as $GNN(A, X)$, where $A$ and $X$ are the adjacency and attribute matrices, respectively. We rescale the adjacency matrix with edge influences $Z$ as follows: $A_Z = Z \odot A$.

The GNN treats $A_Z$ in the same way as the original matrix: $H = GNN(A_Z, X)$

We generate a graph representation $h$ from the node representation matrix $H$ via a mean pooling operator. The graph representation $h$ is then given as input to a classifier that will predict the graph labels $y$. Here, we use an MLP as our classifier.

---

**Algorithm 1** EAGER

---

**Require:** Graphs $A_{1:n}$, node attributes $X_{1:n}$, labels $y_{1:n}$, explainer $g_{\Phi_0}$, number of outer/inner loops $\kappa$ and $T$, threshold parameter $\alpha$.
**Ensure:** Trained $g_{\Phi_\kappa}$
 1: **for** $\tau \in [0, \kappa - 1]$ **do**
 2:  $A^{tr}, A^{sup}, X^{tr}, X^{sup}, y^{tr}, y^{sup} \leftarrow \text{split}(A_{1:n}, X_{1:n}, y_{1:n})$
 3:  $Z^{tr} \leftarrow g_{\Phi_\tau}(A^{tr}, X^{tr})$
 4:  **for** $t \in [0, T-1]$ **do**
 5:    $H^{tr} \leftarrow GNN_{\tau,t}(Z^{tr} \odot A^{tr}, X^{tr})$
 6:    $h^{tr} \leftarrow POOL_{mean}(H^{tr})$
 7:    $p^{tr} \leftarrow MLP_{\tau,t}(h^{tr})$
 8:    $GNN_{\tau,t+1}, MLP_{\tau,t+1} \leftarrow inner\text{-}opt \; f_{Z^{tr}}(p^{tr}, y^{tr})$
 9:  **end for**
10:  $Z^{sup} \leftarrow g_{\Phi_\tau}(A^{sup}, X^{sup})$
11:  $p^{sup} \leftarrow MLP_{\tau,T}(POOL_{mean}(GNN_{\tau,T}(Z^{sup} \odot A^{sup}, X^{sup}))$
12:  $g_{\Phi_{\tau+1}} \leftarrow outer\text{-}opt \; F_{\theta_T}(p^{sup}, y^{sup}, \alpha)$
13: **end for**
14: **return** $g_{\Phi_\kappa}$

---

## 3.4 BILEVEL OPTIMIZATION TRAINING

The main steps performed by our model (EAGER) are given in Algorithm 1. For each outer iteration (lines 1-13), we split the training data into two sets—training and support—(line 2). First, we use training data to calculate $Z^{tr}$, which is used for $GNN$ training in the inner loop (lines 4-9). Then, we apply the gradients from the inner problem to optimize the outer problem using support data (lines 10-12). The main output of our algorithm is the explainer $g_{\Phi_\kappa}$. Moreover, the last trained $GNN_{\theta_T}$ can also be used for the classification of unseen data or a new GNN can be trained based on $Z$. In both cases, the GNN will be trained with the same input graphs. For gradient calculation, we follow the gradient-based approach described in Grefenstette et al. (2019).

### 3.4.1 PREDICTOR TRAINING (INNER LOOP)

At inner loop iterations, we keep gradients while optimizing model parameters $\theta$.

$$\theta_{t+1} = inner\text{-}opt_t \left(\theta_t, \nabla_{\theta_t} \ell^{tr}(\theta_t, Z_\tau)\right)$$

After T iterations, we compute $\theta^*$, which is a function of $\theta_1, \ldots, \theta_T$ and $Z_\tau$, where $\tau$ is the number of iterations for meta-training. Here, $inner\text{-}opt_t$ is the inner optimization process that updates $\theta_t$ at step $t$. If we use SGD as an optimizer, $inner\text{-}opt_t$ will be written as follows with a learning rate $\eta$:

$$inner\text{-}opt_t \left(\theta_t, \nabla_{\theta_t} \ell^{tr}(\theta_t, Z_\tau)\right) := \theta_t - \eta \cdot \nabla_{\theta_t} \ell^{tr}(\theta_t, Z_\tau)$$

In our work, we focus on the graph classification problem. The loss $l^{tr}$ is cross-entropy loss.

### 3.4.2 EXPLAINER TRAINING (OUTER LOOP)

After $T$ inner iterations, the gradient trajectory saved to $\theta^*$ will be used to optimize $\Phi$. We denote $outer\text{-}opt_\tau$ as outer optimization that updates $\Phi_\tau$ at step $\tau$. The meta-training step is written as:

$$\begin{aligned}
\Phi_{\tau+1} &= outer\text{-}opt_\tau \left(\Phi_\tau, \alpha \nabla_{\Phi_\tau} \ell^{sup}(\theta^*)\right) \\
&= outer\text{-}opt_\tau \left(\Phi_\tau, \alpha \nabla_{\Phi_\tau} \ell^{sup}(inner\text{-}opt_T(\theta_T, \nabla_{\theta_T} \ell^{tr}(\theta_T, Z_\tau)))\right),
\end{aligned}$$

where $\alpha$ controls the rate of learning of the explainer $\Phi$, which in turn controls the tradeoff between the compactness of the explanation and the predictive performance. Notice that unlike the hyperparameter $\beta$ from Equation 1, $\alpha$ does not affect information bottleneck directly, but implicitly via the learning rate of the outer optimization. Hence, it is necessary to distinguish them.

After each meta optimization step, we calculate edge influences $Z_{\tau+1}$ using $g_{\Phi_{\tau+1}}(.)$. Our training algorithm is more computationally intensive than training a simple GNN architecture. For that reason, we set $T$ to a small value. Note that both edge influence learning and graph classification use GNNs to produce node and graph representations. These processes can either use separate GNNs or share the same architecture, also known as weight sharing. We adopt the latter.

## 3.5 Time Complexity

Because EAGER relies on bilevel optimization, its time complexity is a critical discussion. Let $C_{GNN}$ be the running cost of the underlying GNN and $C_{hyper}(T, d)$ be the cost of calculating hypergradients, with $T$ being the number of inner iterations and $d$ being the dimension size. The running times of EAGER during training and inference are $O(2C_{GNN})$ and $O((T + 1)C_{GNN} + C_{hyper}(T, d))$, respectively. Notice that, in general, $C_{hyper}(T, d)$ scales linearly with the number of input graphs when the transition function (to go from one inner iteration to the next) is a typical neural network such as GNN Shaban et al. (2019). When $T$ is small, our method is efficient with respect to the running time of the backbone GNN. Appendix D provides details on the running time.

## 4 Experiments

### 4.1 Datasets

**Graph Classification Datasets** We consider 5 binary graph classification datasets from Molecu-leNet (Wu et al., 2018): BBBP, CLINTOX, TOX21, TOXCAST, and HIV. Additionally, we include MUTAGENICITY (Kazius et al., 2005; Debnath et al., 1991; Morris et al., 2020), a popular molecular classification dataset with ground-truth explanations. However, since there is currently no consensus on these explanations (Tan et al., 2022; Luo et al., 2020; Debnath et al., 1991), we exclude MUTAGENICITY from the evaluation of the explanations produced by EAGER and the baselines.

**Constructing Molecular Synthetic Datasets** Most graph-based datasets with ground-truth explanations consist of synthetic non-molecular graphs (Agarwal et al., 2023). Evaluating explainer models using these datasets may not reflect the expected behavior of these models in real-world settings (e.g., drug discovery) due to the disparity between synthetic graphs and real-world graphs. Instead, we curate 3 semi-synthetic molecular graph datasets with ground-truth explanations. We screen millions of molecules from the ChemBL database (Gaulton et al., 2012) and extracted those with either or both Lactam and Benzoyl functional groups. Lactam groups are cyclic amides with various ring sizes and the Benzoyl group is a benzene ring attached to a carbonyl group. The 3 datasets are:

- **LACTAM**: Positive if molecules containing a Lactam group and negative otherwise. There is no molecule with multiple Lactam groups.

- **BENLAC** (Benzoyl Lactam): Positive if molecules containing a Lactam group; negative if molecules containing a Benzoyl group, a Lactam group with a Benzoyl group, or no Lactam and Benzoyl groups. There is no molecule with multiple Lactam or Benzoyl groups.

- **BENLACM** (Benzoyl Lactam Multiclass): Class I if molecules contain a Lactam group; class II if molecules contain a benzoyl group. Each molecule has either a Lactam or a Benzoyl group. We do not term the labels positive or negative since there is no class of interest in this setting.

One favorable aspect of these datasets is that the prediction is simple for most classifiers yet the explanation is not trivial, as we show in Section 4.4 and Section 4.3. This property allows us to assess not only the quality of the explanation, but also the ability of the classifier to pick up the right signals from data. Both aspects are important for EAGER and ante-hoc explainers as both the explanations and the classifier are learned together. More details on these datasets are provided in Appendix A.

### 4.2 Experimental Settings

**Baselines** We compare EAGER against a variety of baselines. For classification, we include classical models such as GCN (Kipf & Welling, 2016), GAT (Veličković et al., 2017), and GIN (Xu et al., 2018). We also include GMT (Baek et al., 2021), a state-of-the-art GNN with attention-based

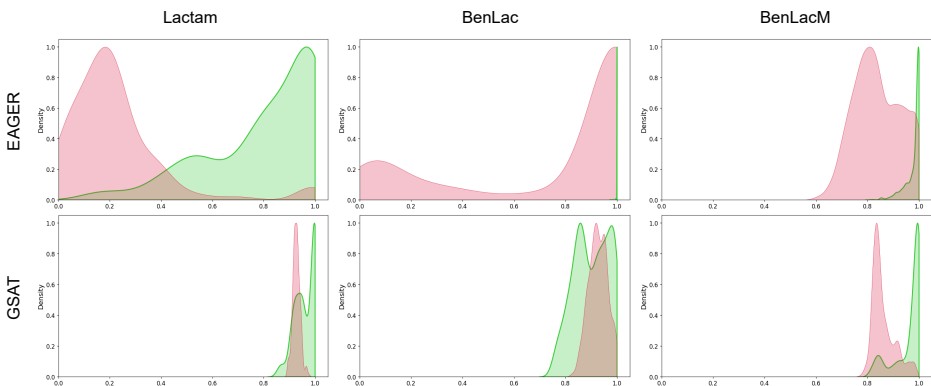

Figure 4: Comparisons between the absolute weights assigned by EAGER and GSAT on ground-truth explanation (foreground) edges and the remaining (background) edges. The plots illustrate the density of edge weights across 3 synthetic molecular datasets. Densities of foreground edge weights are shown in green while those of background edge weights are shown in pink. The density of foreground edges by EAGER on BENLAC is barely visible because most foreground weights are close to 1.0.

Table 1: Performance of EAGER and other baselines on explainability. Based on the ground-truth explanations, we calculate performances in terms of explanation AUC (ExAUC) and precision among the top 10 weighted edges (Precision@10).

| Models | Lactam | | BenLac | | BenLacM | |
|---|---|---|---|---|---|---|
| | ExAUC | Precision@10 | ExAUC | Precision@10 | ExAUC | Precision@10 |
| EAGER (ours) | **98.2 ± 1.6** | 85.9 ± 6.8 | 81.7 ± 7.9 | **87.0 ± 8.4** | **92.9 ± 2.8** | **83.6 ± 3.6** |
| GSAT | 97.7 ± 1.9 | 80.1 ± 6.7 | **84.5 ± 8.3** | 73.5 ± 10.6 | 74.1 ± 7.5 | 71.2 ± 11.5 |
| CAL | 81.2 ± 7.4 | 68.5 ± 12.3 | 69.6 ± 8.1 | 64.8 ± 5.3 | 70.7 ± 5.3 | 55.2 ± 4.3 |
| PGExplainer | 96.9 ± 0.8 | **87.4 ± 4.1** | 73.3 ± 1.9 | 71.6 ± 2.6 | 86.5 ± 1.8 | 81.0 ± 0.1 |
| GNNExplainer | 80.1 ± 1.7 | 50.8 ± 3.4 | 51.7 ± 1.1 | 65.2 ± 6.8 | 57.6 ± 0.8 | 36.4 ± 1.5 |
| GEM | 68.1 ± 11.1 | 48.4 ± 17.9 | 55.0 ± 6.2 | 56.5 ± 8.9 | 56.6 ± 5.8 | 42.0 ± 5.5 |
| CFF | 93.5 ± 0.6 | 81.4 ± 1.3 | 50.7 ± 0.9 | 50.4 ± 2.3 | 51.4 ± 0.8 | 38.9 ± 1.2 |
| OrphicX | 86.8 ± 5.2 | 73.4 ± 8.2 | 78.4 ± 6.3 | 66.4 ± 4.3 | 79.6 ± 7.1 | 63.8 ± 4.7 |

graph pooling, and VIB-GSL (Sun et al., 2022), a method that applies graph structure learning. For explanations on the semi-synthetic molecular datasets, we compare EAGER against both inductive explainers (e.g., PGExplainer (Luo et al., 2020), GEM (Lin et al., 2021), OrphicX Lin et al. (2022)) and transductive explainers (e.g., GNNExplainer (Ying et al., 2019), CFF (Tan et al., 2022)). Most importantly, we compare EAGER against existing ante-hoc graph explainers such as GSAT (Miao et al., 2022), DIR-GNN (Wu et al., 2021), and CAL Sui et al. (2022).

**Models and Training** For GSAT, we follow the configuration suggested by the authors for OGB datasets (Miao et al., 2022). For DIR-GNN (Wu et al., 2021) and VIB-GSL (Sun et al., 2022), we try our best to finetune the models. For any baseline that does not support edge features, we replace their GNN implementation with ours. Specifically, for fair and consistent comparisons, we choose GIN (Xu et al., 2018) with edge encodings as the backbone of all explainers. Following the experiments from previous works (Hu et al., 2020a) on molecular datasets, we construct the GIN backbone with 5 layers and 300 hidden dimensions. We also keep this architectural setting for other GNN baselines, including GCN (Kipf & Welling, 2016), GAT (Veličković et al., 2017), and GMT (Baek et al., 2021). For post-hoc baselines, we obtain their performances via GNN-X-Bench (Kosan et al., 2023). For our experiments on EAGER, we use Adam optimizer, batch size 256, learning rate $1e - 4$, and weight decay 0. All datasets are split with the 8:1:1 ratio for train, validation, and test splits. Experiments are conducted on a shared computing server with random access to either Nvidia V100 or A100 GPUs.

### 4.3 RESULTS ON EXPLAINABILITY

In Table 1, we show the explanation performances of EAGER and other explainers on the 3 semi-synthetic molecular datasets. Two important metrics, explanation AUC and precision among the top

Table 2: Comparing $I(S; G)$ by GSAT and EAGER. For GSAT, we vary the IB thresholding hyperparameter $r$. For EAGER, we vary the multiplicative factor $\alpha$.

| Models | Lactam | BenLac | BenLacM |
|---|---|---|---|
| $\text{GSAT}_{r=0.1}$ | 0.862 ± 0.086 | 0.829 ± 0.031 | 0.636 ± 0.082 |
| $\text{GSAT}_{r=0.3}$ | 0.865 ± 0.071 | 0.836 ± 0.055 | 0.624 ± 0.020 |
| $\text{GSAT}_{r=0.5}$ | 0.871 ± 0.034 | 0.869 ± 0.014 | 0.668 ± 0.073 |
| $\text{GSAT}_{r=0.7}$ | 0.917 ± 0.029 | 0.846 ± 0.037 | **0.615 ± 0.021** |
| $\text{GSAT}_{r=0.9}$ | 0.881 ± 0.061 | 0.857 ± 0.028 | 0.637 ± 0.053 |
| $\text{EAGER}_{\alpha=0.1}$ | 0.792 ± 0.015 | 0.797 ± 0.015 | 0.709 ± 0.015 |
| $\text{EAGER}_{\alpha=0.2}$ | **0.767 ± 0.043** | 0.829 ± 0.022 | 0.678 ± 0.028 |
| $\text{EAGER}_{\alpha=0.5}$ | 0.778 ± 0.017 | 0.831 ± 0.026 | 0.626 ± 0.019 |
| $\text{EAGER}_{\alpha=1.0}$ | 0.843 ± 0.012 | **0.817 ± 0.052** | 0.657 ± 0.020 |
| $\text{EAGER}_{\alpha=5.0}$ | 0.852 ± 0.023 | 0.829 ± 0.009 | 0.672 ± 0.034 |

10 weighted edges, are reported based on the ground-truth subgraphs. We compare EAGER directly with GSAT (Miao et al., 2022), a strong ante-hoc baseline. We also include post-hoc explainers of which explanations are based on the trained GIN models obtained from the graph classification experiment. We use all graphs in calculating explanation AUCs and only use graphs with ground-truth explanations in calculating precisions. Appendix A provides more details on these ground truths.

Across the datasets, EAGER performs competitively both in terms of explanation AUC and precision. Remarkably, on the multiclass dataset BENLACM, our method is the best in both metrics. On LACTAM, EAGER obtains the best explanation AUC and the second best precision while on BENLAC, we achieve the best precision and second best explanation AUC. The results suggest that ante-hoc methods like EAGER and GSAT generally do better than post-hoc alternatives, among which we find PGExplaner to be the most competitive.

Figure 4 visualizes the edge weights assigned by EAGER and GSAT. The green density plots show weights assigned to edges belonging to the ground-truth explanations (foreground edges) while the pink density plots show the same for the remaining edges (background edges). Notice that since the semi-synthetic datasets are highly skewed, the actual density of foreground edges would be much smaller than that of background edges, however, for clarity, we normalize all plots. For GSAT, the weights assigned to the foreground edges and the background edges are quite close to each other, while EAGER tends to assign lower weights to background edges. In BENLAC, EAGER assigns weights close to 1.0 for most foreground edges, resulting in a plot that resembles a vertical line.

To further evaluate the quality of the explanations produced by EAGER, we compare them to those produced by other baselines on the *reproducibility* metrics Kosan et al. (2023). Existing works, especially those on post-hoc models, commonly employ the *fidelity* metrics Yuan et al. (2021). However, *fidelity* requires comparing the predictions using the input graphs versus the predictions using the explanations and is not applicable for an ante-hoc model like EAGER because our predictor follows an explainer and the prediction is based on the explanation. *Reproducibility*, instead, measures how explanations alone can predict class labels. We report the results on this metrics in Appendix B. In general, ante-hoc models like EAGER and GSAT outperform post-hoc models on *reproducibility*.

Finally, we report the estimation of mutual information $I(S; G)$ in the synthetic datasets and compare the minimization of this term between EAGER and GSAT in Table 2. This is an important analysis that has been neglected in studies on interpretability on graphs. For both EAGER and GSAT, we vary the corresponding hyperparameters controlling the compactness. We encode the embeddings of the subgraph $S$ as the weighted average of the edge embeddings produced by the explainer. For graph embeddings, we encode them as unweighted averages of edge embeddings. Similar encodings can be extended to GSAT. We use the slicing method proposed in Fayad & Ibrahim (2024) to efficiently estimate the high-dimensional mutual information $I(S; G)$. The results show that EAGER is better (or as good) at minimizing the mutual information between the subgraphs and the input graphs. This confirms the effectiveness of our proposed approach.

### 4.4 RESULTS ON GRAPH CLASSIFICATION

Table 3 shows the results of EAGER and other baselines on chemical classification benchmarks. For each dataset, we report the mean AUC and error from 10 runs. The best performance on each dataset is bolded and the second best performance is underscored. We do not show these highlights for LACTAM,

Table 3: Classification performances in AUC of EAGER and other baselines.

| Models | Lactam | BenLac | BenLacM | BBBP | ClinTox | Tox21 | ToxCast | HIV | Mutagenicity | Avg AUC | Avg Rank |
|---|---|---|---|---|---|---|---|---|---|---|---|
| EAGER (ours) | 100.0 ± 0.0 | 99.6 ± 0.7 | 100.0 ± 0.0 | 67.1 ± 2.1 | 85.8 ± 4.2 | **74.6 ± 0.7** | 62.1 ± 0.9 | 76.2 ± 1.3 | 87.7 ± 1.6 | **75.58** | **2.42** |
| GSAT | 100.0 ± 0.0 | 99.4 ± 1.6 | 100.0 ± 0.0 | 65.8 ± 2.2 | **87.5 ± 2.7** | 72.9 ± 0.9 | 61.4 ± 0.8 | 76.4 ± 1.2 | 82.6 ± 1.3 | 74.43 | 5.42 |
| DIR-GNN | 86.8 ± 4.7 | 89.7 ± 6.8 | 99.2 ± 0.9 | 64.4 ± 2.9 | 79.5 ± 6.0 | 70.8 ± 0.8 | 61.2 ± 0.6 | 76.0 ± 1.6 | 84.3 ± 1.5 | 72.70 | 7.83 |
| CAL | 99.8 ± 0.2 | 99.7 ± 0.4 | 100.0 ± 0.0 | 66.3 ± 1.9 | 84.4 ± 2.5 | 73.6 ± 1.2 | 62.0 ± 0.9 | 76.7 ± 1.2 | 85.9 ± 1.4 | 74.82 | 4.08 |
| GIN | 100.0 ± 0.0 | 98.1 ± 3.9 | 100.0 ± 0.0 | 65.9 ± 1.9 | 83.5 ± 5.0 | 74.3 ± 1.0 | 61.7 ± 0.7 | 75.8 ± 1.2 | 87.4 ± 1.2 | 74.77 | 4.50 |
| GCN | 100.0 ± 0.0 | 99.3 ± 1.0 | 100.0 ± 0.0 | 64.8 ± 2.8 | 83.4 ± 6.7 | 73.9 ± 0.6 | 62.3 ± 1.0 | 76.2 ± 1.6 | **88.0 ± 1.3** | 74.77 | 4.25 |
| GAT | 100.0 ± 0.0 | 98.3 ± 1.8 | 100.0 ± 0.0 | 65.1 ± 1.3 | 82.6 ± 4.5 | 74.0 ± 0.9 | **63.2 ± 0.8** | 73.4 ± 1.9 | 87.5 ± 1.4 | 74.30 | 4.92 |
| GMT | 100.0 ± 0.0 | 99.5 ± 0.6 | 100.0 ± 0.0 | 65.8 ± 1.9 | 82.3 ± 1.7 | 74.0 ± 0.8 | 62.0 ± 0.7 | 76.8 ± 1.7 | 86.4 ± 1.1 | 74.55 | 4.42 |
| VIB-GSL | 88.2 ± 4.2 | 94.1 ± 4.3 | 99.6 ± 0.4 | **67.5 ± 1.9** | 78.8 ± 2.0 | 71.5 ± 4.9 | 61.1 ± 0.5 | 74.4 ± 1.0 | 83.3 ± 1.4 | 72.77 | 7.17 |

Table 4: Ablation study on varying the GNN backbones and the number of inner epochs. The study is done on chemical classification benchmark with results shown in AUC.

| Models | BBBP | ClinTox | Tox21 | ToxCast |
|---|---|---|---|---|
| $GIN_1$ | 66.8 ± 3.3 | 84.9 ± 4.3 | 72.8 ± 1.1 | 61.0 ± 1.5 |
| $GCN_1$ | 66.5 ± 2.3 | 78.2 ± 5.2 | 74.1 ± 0.7 | 61.9 ± 2.7 |
| $GIN_5$ | 67.1 ± 2.1 | 85.8 ± 4.2 | 74.6 ± 0.7 | 62.1 ± 0.9 |
| $GCN_5$ | 66.2 ± 1.8 | 85.2 ± 4.1 | 74.3 ± 0.6 | 62.5 ± 1.9 |
| $GIN_{10}$ | 66.6 ± 2.4 | 85.7 ± 7.4 | 73.9 ± 1.0 | 62.0 ± 1.3 |
| $GCN_{10}$ | 65.2 ± 1.7 | 82.3 ± 9.2 | 74.6 ± 0.9 | 62.6 ± 1.4 |

BENLAC, and BENLACM because most baselines obtain near-perfect classification on these semi-synthetic datasets. Nevertheless, these datasets are meant for comparing the explanation ability and perfect classification performances promote better comparison of the extracted explanations.

No single baseline obtains the best performance on more than one dataset. EAGER is the best classifier for TOX21 while ante-hoc GSAT (Miao et al., 2022) performs the best on CLINTOX. Substructure learner VIB-GSL (Sun et al., 2022) does particularly well on BBBP but not on other benchmarks. In terms of overall competitiveness, our method EAGER is among the top 2 performing models on 4 out of the 5 reported real-world chemical datasets. Notice that EAGER performs better than its backbone model GIN (Xu et al., 2018) across all chemical benchmarks, confirming the effectiveness of our framework as a classifier.

### 4.5 Ablation Study

**Varying the number of inner epochs and the backbone GNN:** In Table 4, we show the classification results in AUC of multiple EAGER trainings with different backbone GNNs and number of inner iterations. Except for BBBP, EAGER with multiple inner epochs tend to perform better than EAGER with only one inner epoch. Interestingly, more epochs do not guarantee equal or better performances. For example, EAGER with GIN backbone achieves the best AUC on TOX21 using 5 inner epochs. This suggest that the number of inner epochs is an important hyperparameter that should be carefully tuned. There is no clear winner between GIN and GCN as backbone GNNs.

**Learning with and without iterative optimization:** We create a single-level version of EAGER (EAGER-single) to test the effectiveness of our iterative optimization scheme. More specifically, EAGER-single optimizes the explainer and GNN classifier parts in an end-to-end fashion with a single loss function. EAGER consistently and significantly outperforms EAGER-single. We report the detailed results in Appendix E.

### 5 Conclusions

We investigated the problem of generating explanations for GNN-based graph-level classification and proposed EAGER, a novel ante-hoc GNN explainer that supports Information Bottleneck objectives with bilevel optimization. Our adaptation of IB is different from existing literature that typically resorts to variational bounds. We compared EAGER against state-of-the-art graph classification methods and GNN explainers using synthetic and real datasets. The results show that EAGER often outperforms the baselines on multiple accuracy and explainability metrics. We also proposed three new datasets with ground-truth explanations from real-world molecules.

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

# A  Dataset Details

## A.1  Graph Classification Benchmarks

We provide more details about the datasets used in our study. Table 5 and Table 6 show statistics regarding the classification benchmarks and our semi-synthetic datasets, respectively. All benchmarks are medium-size molecular graph datasets with both node features and edge features.

Table 5: Molecular Classification Benchmarks

| Datasets | Number of Graphs | Number of Tasks | Number of Node Features | Number of Edge Features |
|---|---|---|---|---|
| BBBP | 2039 | 1 | 9 | 3 |
| ClinTox | 1478 | 2 | 9 | 3 |
| Tox21 | 7831 | 12 | 9 | 3 |
| ToxCast | 8575 | 617 | 9 | 3 |
| Mutagenicity | 4337 | 1 | 14 | 3 |

Table 6: Semi-synthetic Molecular Benchmarks with Ground-truth Explanations.

| Dataset | Task Type | Graph Type | Number of Graphs | Edge Label | Class |
|---------|-----------|------------|-----------------|------------|-------|
| Lactam | Binary | w lactam groups | 200 | Yes | Positive |
| | | w/o lactam groups | 1000 | No | Negative |
| BenLac | Binary | w lactam groups only | 100 | Yes | Positive |
| | | w benzoyl groups only | 100 | Yes | Negative |
| | | w both lactam and benzoyl groups | 200 | Yes | Negative |
| | | w/o either lactam or benzoyl groups | 800 | No | Negative |
| BenLacM | Multiclass | w lactam groups only | 500 | Yes | 1 |
| | | w benzoyl groups only | 500 | Yes | 2 |

The molecular graph classification datasets reflect real-world applications of graph learning in chemistry. Most of them are obtained from the MoleculeNet (Wu et al., 2018) with featurization following that of Open Graph Benchmark (Hu et al., 2020a). In particular, there are 9 node featuers and 3 edge features describing various atom and bond properties. The Mutagenicity dataset is obtained from TUDataset (Morris et al., 2020). In Mutagenicity, the node features and edge features are one-hot encodings of atom types and bond types. More specific descriptions of the datasets and the tasks are as follows:

- BBBP: Blood-brain barrier permeability.
- ClinTox: Drugs that failed clinical trials for toxicity reasons
- Tox21: Toxicology on 12 biological targets
- ToxCast: Toxicology measurements via high-throughput screening
- Mutagenicity: Ability of an agent to cause genetic mutations

### A.2 SEMI-SYNTHETIC MOLECULAR BENCKMARKS

We introduce 3 semi-synthetic molecular datasets with ground-truth labels (Table 6). Specifically, we screen millions of real molecules from ChemBL (Gaulton et al., 2012) and randomly sample those with Lactam and/or Benzoyl substructures. Both of these functional groups are known to exhibit pharmaceutical values such as anti-bacterial effects. Additionally, lactam groups may have varying sizes, which is interesting as ground-truth explanations. The screening and processing are done via RDKit. Next, we describe the semi-synthetic datasets in more details.

LACTAM is a binary classification dataset to distinguish between molecules containing lactam groups and those not containing the lactam groups, with the former and the latter having the positive and negative labels, respectively. Benzoyl Lactam (BENLAC) is also for binary classification but is a more challenging version of LACTAM. In BENLAC, we consider both the lactam and the benzoyl functional groups. Molecules may contain only lactam groups, only benzoyl groups, both lactam and benzoyl groups, or none of those groups. Out of these molecules, only those containing only lactam groups are the positive class. Benzoyl Lactam Multiclass BENLACM is a multiclass classification dataset. BENLACM requires distinguishing between molecules with either lactam groups or benzoyl groups. In this case, both classes have ground-truth explanations. Since most datasets on explainability consider binary classification with ground-truth explanations only for one class of interest, we believe it is interesting to look into scenarios in which different classes of data possesses different explanations. These datasets have the same node and edge featurizations as those of the MoleculeNet (Wu et al., 2018) benchmarks (e.g BBBP).

## B REPRODUCIBILITY

Reproducibility measures how explanations alone can predict class labels. It is a key property as it allows users to correlate explanations and predictions without neglecting potentially relevant information from the input. We vary the size of the explanations by thresholding edges based on their importance. We then train a GNN using only the explanations and labels. We compare EAGER against post-hoc and ante-hoc explainers on MUTAGENICITY and show the results in Figure 5.

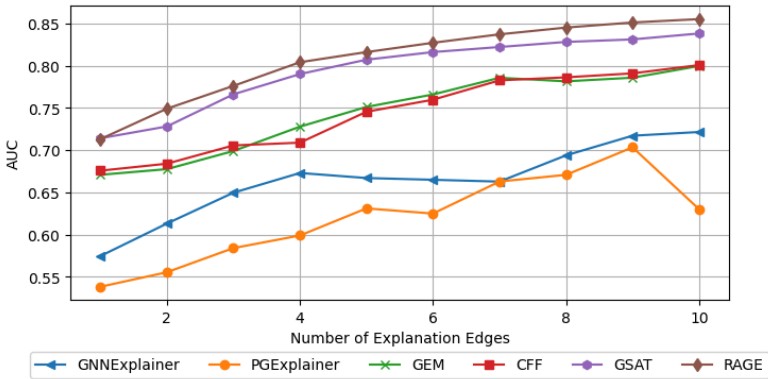

Figure 5: Reproducibility of various ante-hoc and post-hoc explainers on Mutagenicity. We extracted explanation subgraphs by keeping the edges with the highest weights and trained a classifier on the resulting dataset comprising these subgraphs. EAGER and GSAT, being ante-hoc, perform better than other post-hoc explainers under such perturbation.

The results demonstrate that EAGER outperforms competing explainers in terms of reproducibility. Ante-hoc methods, EAGER and GSAT, emerge as the best and second best methods, confirming the superiority of ante-hoc learning in terms of generalization and robustness. EAGER incorporates these qualities through meta-training and bilevel optimization. Two post-hoc explainers, PGExplainer and GNNExplainer, perform poorly. As expected, larger explanations lead to better reproducibility.

## C  LEARNING CURVE

As a method that relies on bilevel optimization, EAGER can suffer from instability during training. In Figure 6, we report the learning curves from the training of EAGER on Mutagenicity. Specifically, the red line and the blue line show progression in the training loss and the validation AUC, respectively. In general, the plots show that the training of EAGER is reasonably stable, with some degree of variation. Despite these variations, the overall trends, reduction in training loss and improvement in validation AUC, are still clearly observed.

## D  RUNNING TIME

Bilevel optimization needs more training time than the standard GNNs. Table 7 shows training and testing times on the LACTAM dataset for all methods. For training, we show, in seconds, the amount of time required to finish one training epoch while for testing, we show the amount of time taken to evaluate the whole test split. We do not include training time for GNNExplainer, GEM, and CFF since these are transductive models. Instead, we report, as testing time, the amount of time it take for these methods to fit and explain every graph in the testing set.

EAGER trains slower than standard and some of the sophisticated GNN methods, while having comparable or faster testing time. For post-hoc graph explainers, EAGER is much faster in testing. For ante-hoc explainers, EAGER is slower than in training than GSAT, is the same in testing compared to GSAT, and is significantly faster than DIR-GNN in both training and testing.

## E  SINGLE-LOOP TRAINING

Table 8 compares learning with EAGER and EAGER-single on several molecular classification datasets. EAGER-single trains both the explainer GNN and the predictor GNN in a sequential end-to-end pipeline without bilevel optimization. EAGER outperforms EAGER-single by a large margin on all tested datasets. This observation confirms the effectiveness of our bilevel optimization strategy.

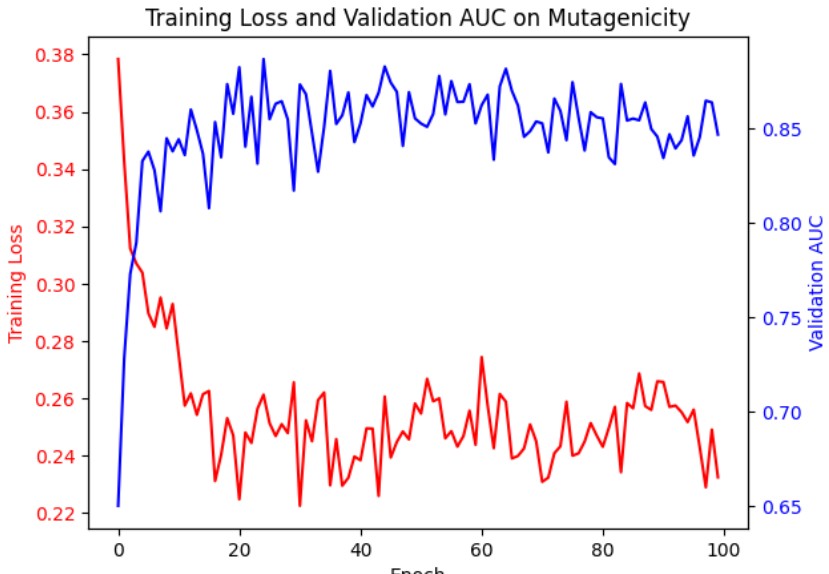

Figure 6: The line plots show the training loss and validation AUC of EAGER on one training fold of Mutagenicity. The training loss is shown in red and the validation AUC is shown in blue.

Table 7: Training and testing time for EAGER and baselines for LACTAM.

|  |  | Training (s/epoch) | Testing (s/fold) |
| --- | --- | --- | --- |
| Standard GNNs | GCN | 0.165s | 0.021s |
|  | GAT | 0.216s | 0.022s |
|  | GIN | 0.203s | 0.021s |
| Sophisticated GNNs | GMT | 0.268s | 0.027s |
|  | VIB-GSL | 40.017s | 7.266s |
| Post-hoc Explainers | PGExplainer | 78.026s | 1.212s |
|  | GNNExplainer | N/A | 111.842s |
|  | GEM | N/A | 88.401s |
|  | CFF | N/A | 2306.452s |
| Ante-hoc Explainers | GSAT | 0.331s | 0.025s |
|  | DIR-GNN | 47.017s | 6.306s |
| Our method | EAGER | 3.252s | 0.025s |

## F  LIMITATIONS

The most critical limitation of EAGER is the training time (Table 7), as we rely on bilevel optimization. Even though our inference time is quite efficient, long training time is still a problem in scenario such as online learning or lifelong learning. We expect this problem to be alleviated by applying more efficient bilevel optimization methods, such as those that apply stochastic samplings, decetralized

Table 8: Ablation study on training with and without bilevel optimization. Classification results are shown in AUC. EAGER with bilevel optimization outperforms EAGER-single in every dataset.

| Models | BBBP | ClinTox | Tox21 | ToxCast |
| --- | --- | --- | --- | --- |
| EAGER (w bilevel optimization) | 67.1 ± 2.1 | 85.8 ± 4.2 | 74.6 ± 0.7 | 62.1 ± 0.9 |
| EAGER-single (w/o bilevel optimization) | 60.0 ± 3.8 | 67.3 ± 6.0 | 65.3 ± 2.4 | 53.3 ± 2.1 |

processing, or single-loop algorithms (Yang et al., 2021; Dagréou et al., 2022; Dong et al., 2023). However, we did not experiment with different meta-learning or bilevel optimization approaches in our study and would like to leave this task for future work.

## G    MORE EXAMPLES OF EXPLANATIONS

We include more examples of the explanations discovered by EAGER. Figure 7, Figure 8, and Figure 9 show examples from LACTAM, BENLAC, and BENLACM, respectively. In these examples, blue highlights indicate ground-truth explanations and red highlights indicate edge importance learned by EAGER. Darker shades of red mean higher weights.

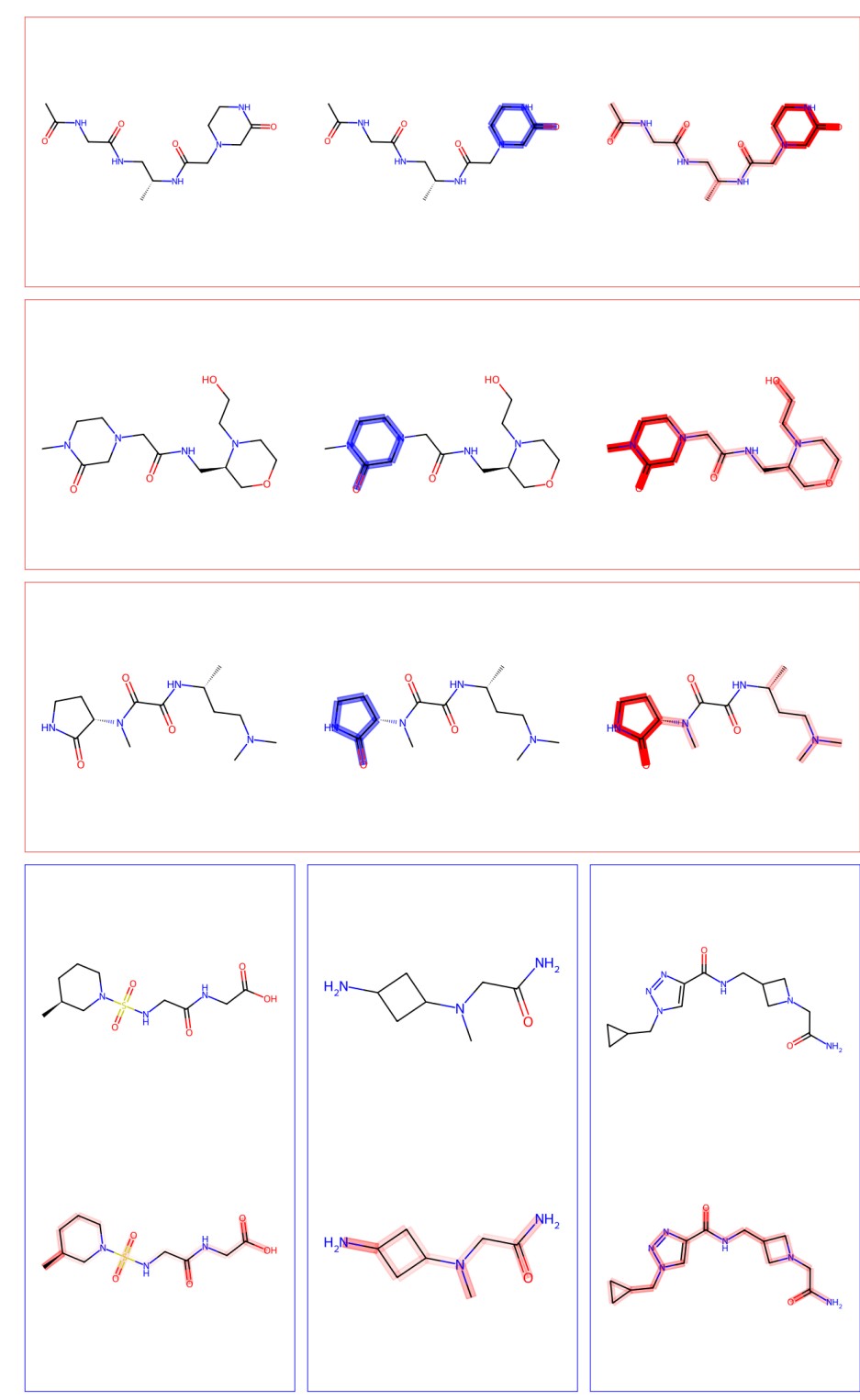

Figure 7: Examples of explanations by EAGER on LACTAM. Positive examples with ground-truth explanations are shown in red boxes and negative examples are shown in blue box.

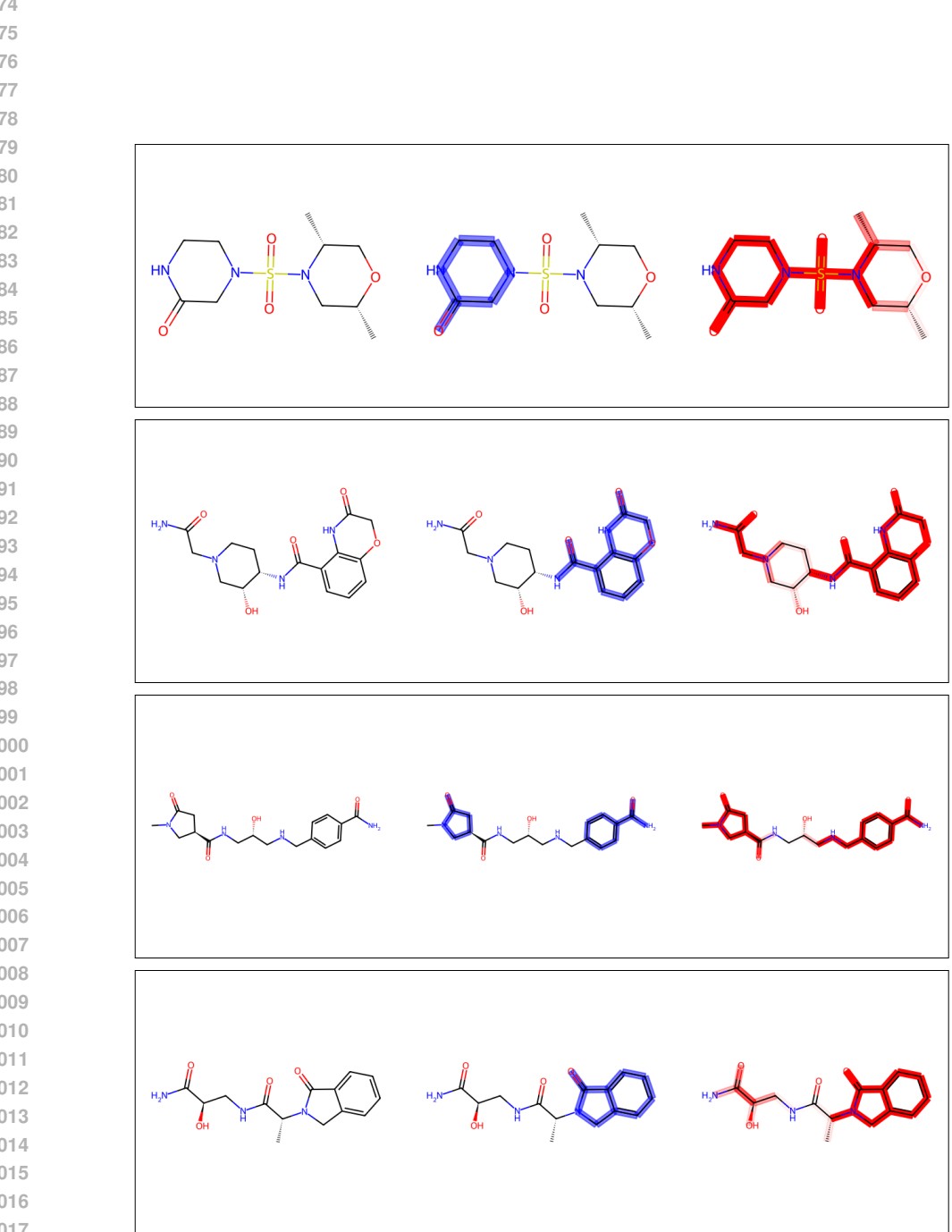

Figure 8: Examples of explanations by EAGER on BENLAC.

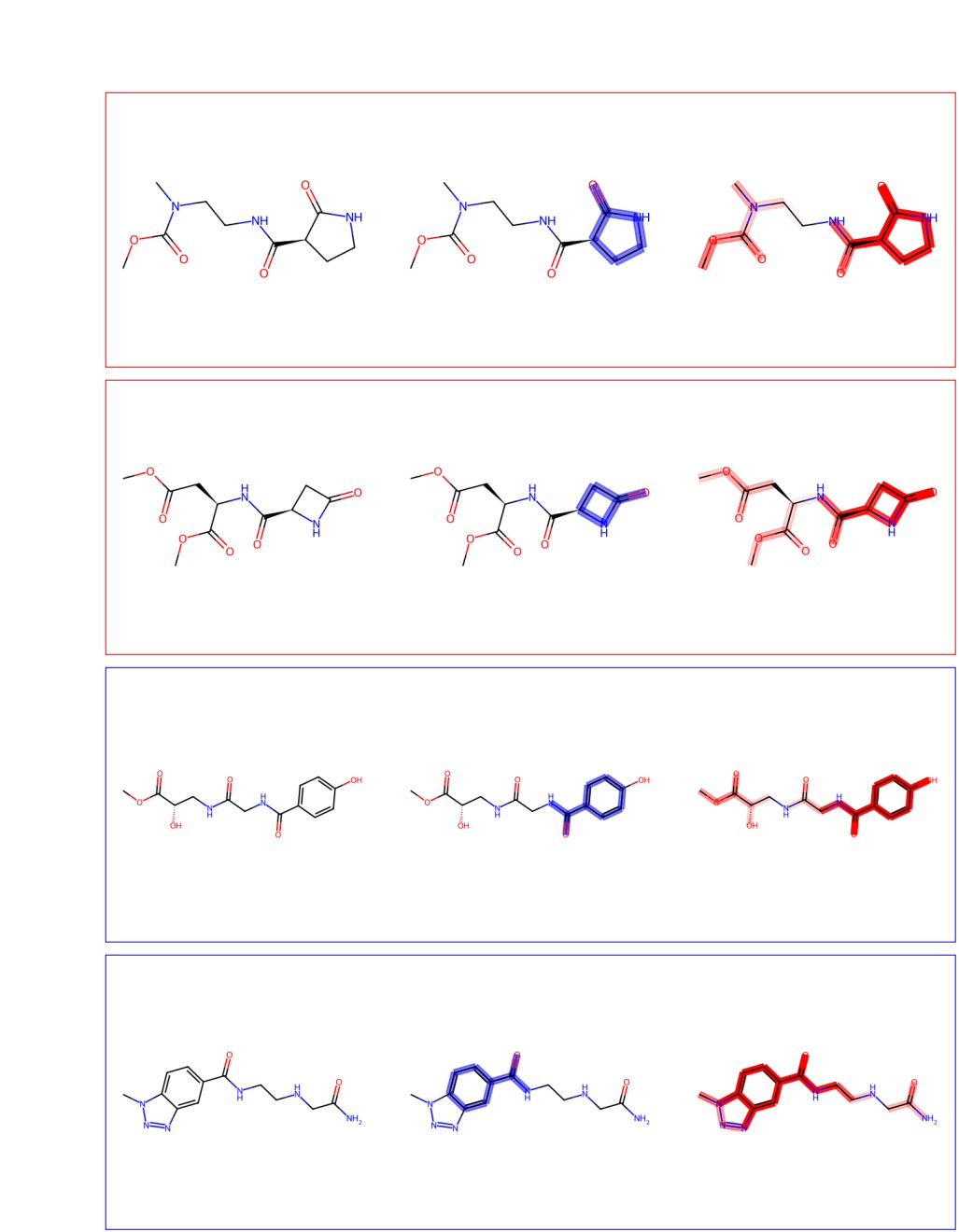

Figure 9: Examples of explanations by EAGER on BENLACM. Red boxes and blue boxes distinguish examples from the 2 classes. Specifically for this dataset, we apply min-max scaling on the weights assigned by EAGER before plotting for better perceptibility.

