# OpenReview forum: "Learning Ante-hoc Explanations for Molecular Graphs"
_ICLR.cc/2025/Conference — Submitted to ICLR 2025_

### Official Review · Reviewer_TsSs · 2024-10-31

**Soundness:** 2
**Presentation:** 1
**Contribution:** 2
**Rating:** 3
**Confidence:** 3

**Summary:**

This paper presents EAGER, a method for producing subgraph explanations for graph neural networks in an *ante hoc* manner. In this method, the input graph is first passed to an explainer network, which learns an edge weight for each edge, where the edge weight reflects the importance/influence of that edge for final prediction. These edge influences are used to modify the input graph (simply scaling the adjacency matrix), which is then passed to the predictor network, which finally predicts the output label. The loss functions are based on information bottlenecking, which uses mutual information to maximize the usefulness of the subgraph explanation for prediction, and minimize the size of the subgraph itself.

In order to train both networks, a meta-learning approach is taken (i.e. bi-level training), where the predictor is trained for several iterations using training data, and the resulting gradients from training the predictor is then used to perform gradient descent on the explainer (using the support dataset).

The EAGER method is based on an existing approach, GSAT, which also uses bi-level training to produce an explainer and predictor network. In contrast with GSAT, which approximates the mutual information between the input graph and the subgraph explanation in the loss using a variational approach, EAGER approximates mutual information using the divergence between their representations.

The experimental results focus on three molecular classification tasks, where the predictive task is to classify molecules with lactam or benzoyl groups. The ground-truth explanations are these lactam or benzoyl groups. The authors show that compared to GSAT (and some *post hoc* explainer methods like PGExplainer), EAGER is able to accurately identify the ground-truth edges in the lactam or benzoyl groups as explanations, and is competitive with other methods or better.

**Strengths:**

### Many references and explanations to previous works

One of the major strengths of this manuscript is the thoroughness when citing other relevant works. I found it very easy to find relevant literature from the citations, and it was easy to understand the contributions of each of those works, even though I don’t have a background in information bottlenecks for interpretability. I also found it easy to understand how this work (EAGER) differs from previous related works (i.e. what the marginal contributions of EAGER are). I wish all papers in AI/ML were this thorough in references and describing what the marginal contributions are.

### Good evaluations of the accuracy of the explanations

For the datasets where the accuracy of the explanations were evaluated, the evaluations were decently thorough. The accuracy of the explanations was shown by measuring the accuracy of edges which were weighted properly. It was also very informative to see the distribution of edge weights given to the proper ground-truth (i.e. lactam or benzoyl) edges compared to the other background edges, comparing EAGER and GSAT. This figure is particularly compelling, in my opinion.

**Weaknesses:**

### Experimental results on explainability are on very easy tasks with identical explanations

The three datasets used in this work to evaluate the accuracy of the explanations are all very easy tasks. All three are based on identifying lactam and/or benzoyl groups in small molecules. The predictive task itself is already extremely simple (a neural network isn’t even needed, technically). More importantly, the correct explanation for every single input graph is going to be the same (i.e. a lactam group or a benzoyl group). That is, there is very little to no variation in the explanations between input graphs.

In contrast, real-world tasks on molecules are likely going to be far more complex (e.g. classify molecules based on solubility or toxicity or drug-like properties). In these real-world tasks, the explanations will be far more diverse compared to the datasets/tasks evaluated here.

Furthermore, the accuracy of explanations from EAGER were only evaluated on these few easy datasets. EAGER is technically a general graph-explainability method, and even though the manuscript is presented as being focused on molecules, it would be very informative to see how it performs on non-molecular graphs. After all, there’s technically nothing that’s preventing EAGER or GSAT from being evaluated on general graphs. Even if this work were to entirely be focused on molecules, it will be crucial to evaluate this method’s performance on more difficult molecular tasks with more diverse explanations. As of now, the predictive tasks are too simple and the correct explanation for every example is the same, which severely limits the evaluation of this method for any reasonable task.

There are other experiments on other molecular datasets, but the results shown are limited to predictive performance, and there are no other results on explainability.

### Unclear details on technical contributions

The writing/flow of the paper is not very clear. The technical details are rather lacking. In particular, the main technical contribution in this paper seems to be the way $I(S, G)$ is calculated in the information-bottleneck loss (paragraph beginning at line 212). However, the exact way this quantity is computed is never really described.

Algorithm 1 is also included to walk through the EAGER algorithm, but it only describes the bi-level meta-learning approach at a high level, and includes the neural-network architecture backbone. It doesn’t sufficiently describe how the loss is computed. Later equations also define the bi-level optimization in terms of the inner and outer loop, but the losses themselves, $\ell^{tr},\ell^{sup}$, are never defined in the paper.

Since the computation of the loss is the major technical novelty of this paper, more details need to be shown describing this development, as well as the previous attempts. Since this paper’s method (EAGER) is most related to GSAT, the related work should describe the variational approach used in GSAT (at least briefly), and many more details should be given for how EAGER is different. The section on bilevel optimization in related works, incidentally, seems not particularly useful.

On a side note, it is not clear what the purpose of Section 3.3.1 is.

### Limited marginal contribution

The marginal technical contribution of this paper seems to be an improvement on GSAT, where one of the terms in the information-bottleneck loss is computed differently (instead of relying on a variational bound). This marginal technical contribution is not huge, but could still be useful if it leads to large improvements overall, or if there are interesting properties (relative to GSAT) stemming from the difference in how the loss is computed.

The marginal empirical contribution would ideally provide evidence of consistent improvements, or experiments showing unique technical insights into the method. However, this paper’s empirical contributions are also a bit limited. There are only a handful of very related and easy tasks evaluated (as mentioned above), which are focused on molecules. Together, both the technical and empirical results are somewhat limited.

### Many grammatical/writing issues

There are also many grammatical issues and other typographical errors throughout the manuscript. These are minor blemishes which are not a big issue, but should be fixed regardless. Here is a *very* non-comprehensive list:

- “The main idea is to find [the] most relevant information” (line 183)
- Equation 2 is missing parentheses in the “exp”
- “two distributions are [kept] constant (line 203)

**Questions:**

### How is $I(S, Y)$ calculated?

The main text says that this is calculated by computing the cross-entropy loss with respect to the labels. Why is the cross entropy a measure of I(S, Y)?

### How is $I(S, G)$ calculated?

The main text mentions an approximation in representation space, but how exactly is this quantity computed?

### How does Equation 2 minimize the objective in Equation 1?

Although Tishby et. al. (2000) proposed this reformulation, it would be great to have some intuition about why these reformulation minimizes the objective function.

### What is the definition of the loss functions $\ell^{tr}$ and $\ell^{sup}$?

The bi-level optimization is key, but the procedure is only described at a very high level (the equations on page 6 only show how the meta-learning is done in general, but not what the losses are. Additionally, what is $\theta^{*}$ exactly?

### How is $\alpha$ related to $\beta$?

Equations 1 and 2 feature the hyperparameter $\beta$, which trades off between predictability and explainability (i.e. compactness of $S$). But Algorithm 1 and Table 2 show $\alpha$ as a hyperparameter (i.e. learning rate), which is meant to do a similar trade-off. What is the relationship between these two hyperparameters? Can Table 2 be replicated to show the same results by tuning $\beta$ instead of $\alpha$?

On a related note, why is $\alpha$ described as a "threshold parameter" in Algorithm 1?

---

> ### Author Response · Authors · 2024-11-20
> **Authors' Rebuttal (1/3)**
>
> Thank you for your time and effort reviewing our paper. We highly appreciate your inputs and would like to address your concerns as follows.
>
> ```Experimental results on explainability are on very easy tasks with identical explanations```
>
> This is a reasonable concern. However, we respectfully disagree with the above statements by the reviewer and would like to clarify these 3 datasets on 3 points:
>
> - The datasets are simple for classification, but not for explanation. Table 3 shows that most predictors can obtain near perfect classification on these 3 datasets, however, if you look at Table 1, the performances on explainability varies significantly. Such observation shows that while all methods excel at prediction, not all of them pick up the right signals. In this case, the ease of getting to the right prediction is a favorable attribute because it sets a fair ground to compare various methods on explainability and robustness to noisy signals, based on the ground-truth explanations. Notice that the point of ante-hoc explanation is learning both the good explanations and good predictions at the same time.
>
> - The ground-truth explanations are not simple. In fact they are quite diverse. We encourage the reviewer to look at Figures 7, 8, and 9 in the appendix for some examples. For instance, Lactam groups can have varying ring sizes and configuration within a molecule (Figure 7). Various combinations of both Lactam groups and Benzoyl groups lead to diverse ground truth explanations in molecules (Figures 8 and 9). This makes our datasets more diverse in terms of explanations compared to existing datasets with house or cycle motifs. Moreover, our graphs are real-world molecules mined from ChemBl, not random graphs.
>
> - We have various prediction settings, not just binary classification on whether the motifs exist. In the dataset BenLac, we assign labels according to various co-occurrence conditions of either the lactam and benzoyl group. Recognizing these logics is beyond a simple pattern matching problem. In the dataset BenLacM, we consider the explanability in a multiclass setting in which each class has a separate pattern. This setting is often overlooked in the literature. Our dataset Lactam is a binary classication task, however, in figure 7, we showed the weight assigned on both the positive and the negative examples, which is often overlooked in the literature.
>
> Certainly, our synthetic datasets cannot capture all the complex interactions that may happen in Chemistry. However, as we show in our Table 2, many existing graph explainers fail to capture even these patterns. We believe these datasets are more complex than many existing benchmark graph datasets for explainability.

---

> > ### Author Response · Authors · 2024-11-20
> > **Authors' Rebuttal (2/3)**
> >
> > ```Unclear details on technical contributions: the exact way [I(S,G)] is computed is never really described, [Algorithm 1] doesn’t sufficiently describe how the loss is computed. How is H(S,Y), H(S,G) calculated?, etc```
> >
> > We thank the reviewers for bringing up these concerns. We acknowledge that the description of the technical details should have been clearer.
> >
> > Firstly, we would like to state that the loss is not the main contribution of the paper. The main contribution of the paper is solving the explainability problem via the IB principle, by aligning the iterative process proposed by Tishby (2000) [1] to a bilevel optimization framework. This iterative process solves the IB objective (Equation 1), however, it is intractable in the graph space and high-dimensional space and thus has never been applied in this domain (or rarely in general in modern deep learning setting). By aligning it to a bilevel optimization framework and utilizing neural parameterization, we offer a way to estimate and execute this iterative algorithm in the graph space.
> >
> > More details below.
> >
> > The iterative process has 2 main parts. The first part (2nd and 3rd lines of Equation 2) estimates the current $P(Y|S)$ given the data and current explanations. The second part (1st line of Equation 2) calculates the $P(S|G)$ that optimizes the IB-objective (Equation 1) given the current estimate of $P(Y|S)$. The formula of the 1st equation is obtained by taking the derivative of the IB-objective with respect to $P(S|G)$. For more details on this derivation, please refer to Tishby (2000) [1].
> >
> > Notice that estimating $P(Y|S)$ can simply be done via a predictive network that minimizes the cross-entropy between $P(Y|S)$ and $P(Y|G)$. This is the common approach performed by EAGER and existing works, such as GSAT and GIB.
> >
> > Now, finding the optimal $P(S|G)$ to minimize $I(S,G)$ is the challenging part. Existing works do this directly via a variational bound on $I(S,G)$. For us, we look at replicating the first equation. Notice that in the 1st line of Equation 2, the value of $P(S|G)$ is adjusted iteratively according to the current divergence between $P(Y|S)$ and $P(Y|G)$. We actually have access to the divergence value from the other steps (2nd and 3rd lines) via calculating the cross-entropy between P(Y|S) and P(Y|G). Cross-entropy and KL-divergence are highly-related values. One can express one in terms of the other.
> >
> > If we set up a bilevel optimization problem where the inner loop performs the steps in the 2nd and 3rd lines, and the outer loop perform the 1st line of Equation 2, then we can replicate the iterative algorithm. The inner loop learns a predictor, optimizing the cross-entropy loss between $P(Y|S)$ and $P(Y|G)$. The outer loop learns an explainer ($P(S|G)$) that get updated via the hypergradients from the inner loop. In this case, the hypergradient is with respect to the inner opimized cross-entropy loss, which, minimized the KL-divergence between $P(Y|S)$ and $P(Y|G)$.
> >
> > The loss is cross-entropy loss. We have updated this information in several other places to prevent further confusion in the future.
> >
> > ```It is not clear what the purpose of Section 3.3.1 is.```
> >
> > Section 3.3.1 is a part of Section 3.3 as a whole where we describe the architectural components of the model, in which 3.3.1 describes the Explainer module and 3.3.2 describes the predictor module.
> >
> > [1] Tishby, Naftali, Fernando C. Pereira, and William Bialek. "The information bottleneck method." arXiv preprint physics/0004057 (2000).

---

> > > ### Author Response · Authors · 2024-11-20
> > > **Authors' Rebuttal (3/3)**
> > >
> > > ```How does Equation 2 minimize the objective in Equation 1?```
> > >
> > > The iterative process (Equation 2) has 2 main parts. The first part (2nd and 3rd lines) estimates the current $P(Y|S)$ given the data and current explanations. The second part (1st line) is a closed-form solution to the IB objective (Equation 1) given the current estimate of $P(Y|S)$. The formula of the 1st equation is obtained by taking the derivative of the IB-objective with respect to $P(S|G)$.
> > >
> > > We have updated the expression of Equation 2 and fixed the explanation of this part (Line 215-217).
> > >
> > > ```What is the definition of the loss functions $l_{tr}$ and $l_{sup}$? The bi-level optimization is key, but the procedure is only described at a very high level (the equations on page 6 only show how the meta-learning is done in general, but not what the losses are. Additionally, what is $\theta *$ exactly?```
> > >
> > > In the context of bilevel optimization, we use different data splits for optimizing the inner problem and the outer problem. Specifically, we split a data batch into a training batch (**tr**) and a support batch (**sup**) for the inner and outer problems, respectively. The losses, $l_{tr}$ and $l_{sup}$, are the objectives of the inner and outer problems. In our case, they both measure classification performance, i.e, cross-entropy loss.
> > >
> > > $\theta *$ represents the solution to the inner problem, which, in this case, is the T-step optimization of the weights of the predictor GNN given the explanations from the outer model (the explainer). If you look at line 318-319, from the first line to the second line, $\theta *$ is re-expressed as $inner-opt$. We hope the explanation clears the confusion.
> > >
> > > ```How is alpha related to beta?```
> > >
> > > In the IB objective (Equation 1), $\beta$ explicitly controls the amount of information bottleneck. However, in the bilevel optimization, we do not have a way to directly control this quantity, so we introduce another parameter $\alpha$ that will implicitly influence the bottleneck. If we look at the first line of the iterative process (Equation 2), $\beta$ controls the amount of adjustment to $P(S|G)$ based on the divergence between $P(Y|G)$ and $P(Y|S)$. This is similar to the learning rate in SGD-based learning.
> > >
> > > However, when we look at Table 2, we can see that $I(S,G)$ does not always show a linear trend with $\alpha$. There's always an optimal value somewhere in the middle that minimize $I(S,G)$. This leads us to think of another interpretation of the purpose of bottlenecking: retaining useful information, i.e, controlling overfitting. Low $\beta$ prioritizes informativeness over compression, which may lead to overfitting, and vice versa. Therefore, the linear trend of bottlenecking moves along the direction of optimizing overfitting, instead of the value of $\alpha$. This may explain why some middle value of $\alpha$ resulted in a lower bottleneck. At this point implicitly control the information bottleneck is still an open question for us and future projects.
> > >
> > > We have updated the main text to clarify more about the relationship between $\alpha$ and $\beta$.

---

> > > > ### Author Response · Authors · 2024-11-25
> > > > **Friendly reminder**
> > > >
> > > > Dear reviewer,
> > > >
> > > > The rebuttal period is ending soon. We would like to hear back from you whether our respond has resolved all of your concerns. If you have any further questions, we are happy to answer.
> > > >
> > > > Authors

---

> > > > > ### Comment · Reviewer_TsSs · 2024-11-25
> > > > > **Response to rebuttal**
> > > > >
> > > > > Thank you to the authors for their response.
> > > > >
> > > > > ### Limited complexity of datasets and explanations
> > > > >
> > > > > This was (and unfortunately remains) my main concern for this submission. Although I appreciate the exploration of EAGER's performance on these datasets (e.g. the co-occurrence analysis, and the weights for both positive and negative examples), the method is only being demonstrated on these very few, simple, and related datasets.
> > > > >
> > > > > In the real world of computational chemistry, these tasks are merely toy examples which are unrealistic and would never be done in this way. A computational chemist who wants to classify/identify lactam rings would just use RDKit. I highly doubt anyone would train a full GNN to classify lactam molecules when one can get 100% accuracy with a few lines of RDKit calls.
> > > > >
> > > > > It is certainly promising to see that EAGER is performing better (in terms of explanations) compared to some other methods, but this is only on these unrealistically simplistic tasks. This paper would be a lot stronger if it included more realistic tasks that people do rely on deep learning for (e.g. mutagenicity, toxicity, solubility, etc.). These also are tasks where the explanations are more diverse, rather than consistently a single lactam ring (maybe of a different size).
> > > > >
> > > > > ### Clarifying the details of the technical contributions
> > > > >
> > > > > The given explanation is very helpful! It still took me some time to get a better intuition on what is going on mathematically, but it may be because I am not as familiar with Tishby et. al.. I hope that in future versions of the manuscript, this level of detailed explanation is included in the main text of the paper.

---

> > > > > > ### Author Response · Authors · 2024-11-25
> > > > > > **Thank you for your response.**
> > > > > >
> > > > > > Dear reviewer,
> > > > > >
> > > > > > Thank you for your response. Regarding your concern about the limited complexity of the dataset, we would like to provide further justification. In short, our datasets are the most complex chemical datasets with ground-truth explanations out there.
> > > > > >
> > > > > > ```In the real world of computational chemistry, these tasks are merely toy examples which are unrealistic and would never be done in this way. A computational chemist who wants to classify/identify lactam rings would just use RDKit. I highly doubt anyone would train a full GNN to classify lactam molecules when one can get 100% accuracy with a few lines of RDKit calls ```
> > > > > >
> > > > > > We believe this point is problematic. We can make the same argument about any explanation datasets out there (Mutagenticy, BA-shapes). We can use RDkit to classify graphs from the Mutagenicity dataset if we know the ground-truth motifs (-NO2, -NH2). We can also use any graph matching algorithm to classify BA-shapes if we know the ground truth subgraphs (houses, cycles, star, etc). In reality, we cannot use RDKit calls to classify because we presumably do not know the ground-truth explanations. In fact, we can say that all benchmark datasets used for graph explanation are "toy datasets", with reasons followed.
> > > > > >
> > > > > > ```It is certainly promising to see that EAGER is performing better (in terms of explanations) compared to some other methods, but this is only on these unrealistically simplistic tasks. This paper would be a lot stronger if it included more realistic tasks that people do rely on deep learning for (e.g. mutagenicity, toxicity, solubility, etc.).```
> > > > > >
> > > > > > You are correct that chemistry is highly complex. As the result, ground truth explanations on chemical data is extremely rare. As far as we know, among chemical benchmark, only MUTAG and Mutagenicity datasets have ground truth explanations. However, even these explanations are very simple and small (-NO2, -NH2 groups), and are still uncertain depending on the publication sources [1][2][3]. As a matter of fact, most of existing work relies on fully synthetic Bernoulli datasets that are not chemical (BA-Shapes, etc). The explanatory motifs (houses, cycles, stars, etc) on these datasets are quite simple compared to our datasets.
> > > > > >
> > > > > > We would like to point out the our proposed datasets, though still simpler than real-life chemistry, is in the right direction of pushing the complexity of benchmark of explanations on molecular graphs. This exactly addresses your concerns. At our best knowledge, there is no other more complex chemical explanatory benchmarks than our datasets.
> > > > > >
> > > > > > Preparing ground-truth explanations for chemical processes is extremely time consuming and requires high-level of domain expertise. We believe these tasks deserve their own projects and is beyond the scope of our conference submission.
> > > > > >
> > > > > > [1] Asim Kumar Debnath, Rosa L Lopez de Compadre, Gargi Debnath, Alan J Shusterman, and Corwin
> > > > > > Hansch. Structure-activity relationship of mutagenic aromatic and heteroaromatic nitro compounds.
> > > > > > correlation with molecular orbital energies and hydrophobicity. Journal of medicinal chemistry, 34
> > > > > > (2):786–797, 1991.
> > > > > >
> > > > > > [2] Dongsheng Luo, Wei Cheng, Dongkuan Xu, Wenchao Yu, Bo Zong, Haifeng Chen, and Xiang Zhang.
> > > > > > Parameterized explainer for graph neural network. arXiv preprint arXiv:2011.04573, 2020.
> > > > > >
> > > > > > [3] Juntao Tan, Shijie Geng, Zuohui Fu, Yingqiang Ge, Shuyuan Xu, Yunqi Li, and Yongfeng Zhang.
> > > > > > Learning and evaluating graph neural network explanations based on counterfactual and factual
> > > > > > reasoning. In Proceedings of the ACM Web Conference 2022, pp. 1018–1027, 2022.

---

> ### Author Response · Authors · 2024-11-29
> **Looking forward to your further response**
>
> Dear reviewer,
>
> We have resolved your concern regarding the paper's technical novelty and contribution. We also have responded to your last concern regarding the datasets and would like to know if it has been addressed. As the extended deadline of the rebuttal phrase is approaching, we look forward to your confirmation.
>
> Best regards,
>
> Authors

---

### Official Review · Reviewer_RDQB · 2024-11-04

**Soundness:** 2
**Presentation:** 2
**Contribution:** 1
**Rating:** 3
**Confidence:** 4

**Summary:**

The paper introduces EAGER, an ante-hoc graph explainer that generates interpretable explanations for graph neural network (GNN) predictions. EAGER uses the Information Bottleneck (IB) principle within a bilevel optimization framework to learn compact, discriminative subgraphs that are closely tied to the model’s prediction. In the process, EAGER assigns influence values to edges, which are incorporated into the graph to create an influence-weighted GNN. This approach ensures that the explanations are jointly learned with the model, providing consistent and reproducible insights into the model's decision-making.

**Strengths:**

1. This ante-hoc approach avoids the limitations of post-hoc explainers, which often provide inconsistent explanations due to their black-box nature.

2. The paper incorporates edge features directly into the explanation process, which is particularly beneficial for domains like molecular graphs, where edge information is critical.

**Weaknesses:**

1. Lack of Novelty.
This paper primarily consists of a combination of methods from other studies. The model’s unique methodology is not clearly emphasized. For example, in the Information Bottleneck principle, the iterative algorithm from [1] is used as-is. Moreover, in the explainer and predictor sections, except for simple tricks like permutation invariance, the method of PGExplainer [2] is used directly.

- [1] Tishby, Naftali, Fernando C. Pereira, and William Bialek. "The information bottleneck method." arXiv preprint physics/0004057 (2000).
- [2] Luo, Dongsheng, et al. "Parameterized explainer for graph neural network." Advances in neural information processing systems 33 (2020): 19620-19631.

2. Lack of Distinction from Existing Ante-Hoc Models.
The paper does not present advantages that differentiate it from existing ante-hoc models. For example, it does not explain how the bilevel training approach provides any benefits over GSAT, which uses variational bounds. Furthermore, it lacks an explanation of advantages compared to other GNN models that generate predictions and explanations simultaneously, such as CAL [3] and OrphicX [4].

- [3] Sui, Yongduo, et al. "Causal attention for interpretable and generalizable graph classification." Proceedings of the 28th ACM SIGKDD Conference on Knowledge Discovery and Data Mining. 2022.
- [4] Lin, Wanyu, et al. "Orphicx: A causality-inspired latent variable model for interpreting graph neural networks." Proceedings of the IEEE/CVF Conference on Computer Vision and Pattern Recognition. 2022.

3. Need for Fidelity Score in Explanation Evaluation.
In addition to calculating explanation AUC, it is necessary to utilize the Fidelity score [5], which is widely used. It is recommended to assess explanations based on the difference in predicted labels between graphs with and without explanations.

- [5] Yuan, Hao, et al. "On explainability of graph neural networks via subgraph explorations." International conference on machine learning. PMLR, 2021.

4. Limited Baselines.
The baselines in this paper are relatively limited in terms of the explainer models used for comparison.
CAL [6] and OrphicX [7] are models that predict labels based on important explanatory subgraphs. It would be beneficial to include these as additional baselines for both explanation and classification performance.

- [6] Sui, Yongduo, et al. "Causal attention for interpretable and generalizable graph classification." Proceedings of the 28th ACM SIGKDD Conference on Knowledge Discovery and Data Mining. 2022.
- [7] Lin, Wanyu, et al. "Orphicx: A causality-inspired latent variable model for interpreting graph neural networks." Proceedings of the IEEE/CVF Conference on Computer Vision and Pattern Recognition. 2022.

**Questions:**

Since it uses bilevel optimization, learning might be unstable. Could you show the training curve for loss and accuracy?

---

> ### Author Response · Authors · 2024-11-20
> **Authors' Rebuttal (1/2)**
>
> Thank you for your time reviewing our paper. We would like to address your concerns as follows.
>
> ```Lack of Novelty. This paper primarily consists of a combination of methods from other studies. The model’s unique methodology is not clearly emphasized. For example, in the Information Bottleneck principle, the iterative algorithm from Tishby (2000) [1] is used as-is.```
>
> We respectfully disagree with the assertion that our method is merely a combination of approaches from existing studies. While our work draws inspiration from established principles, the development of our framework is far from trivial.
>
> Firstly, the iterative procedure proposed by Tishby (2000) [1] is not applied as is, as doing so in the graph space would be intractable. This is a major challenge as even though the IB principle has been applied to explainability on graphs in many existing works, we are the first to approach the problem via this iterative principle. This constitutes our first key contribution.
>
> Secondly, our contribution lies in aligning this iterative process within a bilevel optimization framework. Specifically, we replace the estimation of $P(S|G)$ and $P(Y|S)$ with neural approximation, achieved through a predictor trained in the inner loop using cross-entropy loss and an explainer trained in the outer loop via hypergradients from the inner loop. In our case, cross-entropy indirectly influences the optimization of mutual information, which is a distinction of our method compared with existing approaches that approximate mutual information via variational bounds. This design enables the predictor to generalize to unseen data, effectively transforming the entire framework into an ante-hoc predictive model. These contributions require deliberate and non-trivial insights.
>
> ```Moreover, in the explainer and predictor sections, except for simple tricks like permutation invariance, the method of PGExplainer [2] is used directly.```
>
> Firstly, PGExplainer [2] is a post-hoc model, whereas our approach is an ante-hoc model. This distinction fundamentally affects how the models are trained and utilized.
>
> Secondly, the structural similarity between our explainer and predictor modules and those of PGExplainer arises from the intuitive and practical design of using an edge-weighting module followed by a prediction module for weighted graphs. The novelty of our method lies elsewhere. Specifically, our approach introduces a bilevel optimization framework inspired by iterative IB principles. This innovative training formulation represents a significant departure from PGExplainer's methodology.
>
> We opted for straightforward design choices for the submodules to focus on showcasing the advantages of our training framework. Our framework is flexible: if needed, it can accommodate more general explainer modules and sophisticated predictors without compromising its core functionality.
>
> [1] Tishby, Naftali, Fernando C. Pereira, and William Bialek. "The information bottleneck method." arXiv preprint physics/0004057 (2000).
>
> [2] Luo, Dongsheng, et al. "Parameterized explainer for graph neural network." Advances in neural information processing systems 33 (2020): 19620-19631.

---

> ### Author Response · Authors · 2024-11-20
> **Authors' Rebuttal (2/2)**
>
> ```Lack of Distinction from Existing Ante-Hoc Models. The paper does not present advantages that differentiate it from existing ante-hoc models. For example, it does not explain how the bilevel training approach provides any benefits over GSAT, which uses variational bounds. Furthermore, it lacks an explanation of advantages compared to other GNN models that generate predictions and explanations simultaneously, such as CAL and OrphicX.```
>
> We could have been clearer on presenting the advantages of EAGER over existing ante-hoc models. Information Bottleneck (IB) principle has been a great basis for explainability. In the graph domain, methods that rely on IB principle often approximate the mutual information via variational bounds, such as GSAT. In order to minimize $I(S;G)$, GSAT uses variational bound $I(S;G) \leq E_{G} [KL(P(S|G) || Q(S))]$, and minimizes the RHS, thus effectively minimizing an upper bound. This can lead to a loose approximation due to the complexity of the graph space. In particular, defining an appropriate variational distribution is difficult in the graph space and one has to make simplifying assumptions regarding features and edge independence. This overhead design is another significant burden. For EAGER, we decouple learning the explainer and the classifier by formulating the learning as a bilevel optimization problem. More specifically, EAGER is inpired by the IB iterative process (cite the paper) that optimizes the IB objective, guaranteeing convergence to local optima. In EAGER, cross-entropy loss indirectly influences the optimization of mutual information.
>
> In terms of the input to the classifiers, EAGER is deterministic as the weighted graph produced by the explainer is used for prediction, taking advantage of modern GNN's ability to process edge features. The weighted graphs produced by EAGER explainer often has highly contrastive distintion between the foreground (high weighted edges) and the background, making the explanation highly interpretable (see Figures 1, 4, and 7-9). GSAT, instead, is stochastic as the method relies on generating random subgraphs based on the distribution represented by the weighted graph. This sampling process favors more uniform distribution to ensure well-behaved gradients, which explains the small difference between foreground and background edges in GSAT's explanations (Figure 4). This means that GSAT explanations are often not sparse and, given that ground-truth explanations are often not available in real-world application, GSAT explanations hard to interpret
>
> Thank you for introducing causality-based methods like CAL and OrphicX . These works are highly related and we have included more discussions about them in our main text (Line 45-56, 135-139). Compared to EAGER or IB-based methods in general, CAL has more assumptions about the existence of causal and shortcut features. Additionally, modeling these features add to the complexity of designing and training the model. OrphicX is a post-hoc model because the target GNN is pretrained and fixed.
>
> ```Need for Fidelity Score in Explanation Evaluation...```
>
> Fidelity is not relevant for EAGER or ante-hoc explanations in general because we learn both the explainer and the classifier at the same time, not adapting an explainer to an already trained classifier in post-hoc explanation. In ante-hoc explanation, both in graph and other domains, the explaner is learned together with the classifier and is part of the system. As such, producing a prediction requires both the explainer and the classifier. Fidelity requires comparing the differences between predictions using the input graphs and predictions using the explanations. However, in our case, the predictor never predicts based on the input graphs so this metric is not applicable.
>
> Instead, we further evaluated the quality of the explanation using a related metric called Reproducibility, which compares how quickly predictive performances drop as we progressively sparsify the explanations. This is reported in Appendix B. We have updated the main text to better present this point.
>
> ```Limited baselines...CAL and OrphicX are models that predict labels based on important explanatory subgraphs. It would be beneficial to include these as additional baselines for both explanation and classification performance.```
>
> As suggested, we have added these baselines. The reviewer can find results for CAL in Table 1 and Table 3, and the results for OrphicX in Table 1. Since OrphicX is a post-hoc model, we did not include it in Table 3.
>
> ```...Could you show the training curve for loss and accuracy?```
>
> We have updated the appendix with this analysis. Please refer to the newly added appendix C. The plots show some variations in terms of training loss and validation AUC. Overall, we believe such level of variations is reasonable small.  The general trends, reduction in training loss and improvement in validation AUC, are still clearly observed.

---

> > ### Author Response · Authors · 2024-11-25
> > **Friendly reminder**
> >
> > Dear reviewer,
> >
> > The end of the rebuttal period is coming soon. We would like to hear back from you whether our respond has resolved all of your concerns and are happy to answer any remaining questions.
> >
> > Authors

---

> > > ### Author Response · Authors · 2024-11-29
> > > **Looking forward for your response.**
> > >
> > > Dear reviewer,
> > >
> > > We have not heard back from you. As the deadline of the rebuttal is approaching soon, we are looking forward to your response. We would like to know if your concerns have been addressed and are happy to have any further discussions.
> > >
> > > Best regards,
> > >
> > > Authors

---

### Official Review · Reviewer_L7oc · 2024-11-04

**Soundness:** 4
**Presentation:** 4
**Contribution:** 3
**Rating:** 8
**Confidence:** 5

**Summary:**

The paper proposes EAGER (Effective Ante-hoc Graph Explainer), an innovative framework designed to produce explainable predictions in graph neural networks (GNNs), particularly for molecular classification tasks. By utilizing the Information Bottleneck (IB) principle and bilevel optimization, EAGER jointly learns a GNN and its explainer, producing both accurate and interpretable predictions. The authors present competitive results across various datasets, demonstrating EAGER's superior performance compared to both ante-hoc and post-hoc explainers.

**Strengths:**

1. Introduces a novel ante-hoc approach that optimizes explainability alongside prediction, addressing limitations of post-hoc methods.

2. Successfully applies a theoretically sound adaptation of the Information Bottleneck principle within GNNs for robust feature selection.

3. Shows empirical advantages over baselines in accuracy, explainability, and reproducibility across synthetic and real-world datasets.

4. Offers substantial evaluation, including interpretability benchmarks, ablation studies, and reproducibility analyses.

**Weaknesses:**

1. Complex Training Process: The bilevel optimization, though effective, is computationally intensive and requires significant training time compared to other models.

2. Limited Practical Validation: EAGER’s application is restricted to curated datasets; more real-world, large-scale evaluations could better demonstrate its adaptability.

3. Reliance on Specific Hyperparameters: Model performance is sensitive to hyperparameter settings, notably in the inner and outer loop parameters of bilevel optimization.

4. Interpretability Metrics: Just for suggestion, it would be better to have more real-world datasets. For those lacking a ground truth explanation, the fidelity score could be considered.

**Questions:**

1. Could you elaborate on the rationale for including the average AUC in Table 3? Is averaging the model’s performance across diverse datasets meaningful or informative in this context?

2. Are there plans to include newer baselines in future evaluations? For instance, the addition of MixupExplainer (2023) might provide useful insights for comparing EAGER's performance with recent advances in explainability.

---

> ### Author Response · Authors · 2024-11-20
>
> Thank you for reviewing our paper. We are happy to answer your questions.
>
> ```The bilevel optimization, though effective, is computationally intensive and requires significant training time compared to other models.```
>
> We included the runtime analysis of EAGER and other baselines in Appendix D-Table 7. EAGER's training time is considerably long, however, we are still significantly faster than several other baselines. EAGER is just as fast as other methods during inference. That said, we believe that EAGER can be sped-up using recent advancements in faster bilevel optimization solvers, such as methods that apply decentralized processing, or single-loop algorithms. We also added a new section 3.5 discussing the runtime complexity of EAGER.
>
> ```EAGER’s application is restricted to curated datasets; more real-world, large-scale evaluations could better demonstrate its adaptability.```
>
> During this project, we realized that many existing benchmark datasets for explanation  (i.e, Bernoulli graphs with attached house of cycle motifs) are not reflective of real-world settings. Motivated by this problem, we curated datasets that are close to real-world settings.
>
> Specifically, our datasets are real molecules mined from the ChemBl database. The ground-truth explanations are not simple and are quite diverse. We encourage the reviewer to look at Figures 7, 8, and 9 in the appendix for some examples. For instance, Lactam groups can have varying ring sizes and configuration within a molecule (Figure 7). Various combinations of both Lactam groups and Benzoyl groups lead to diverse ground truth explanations in molecules (Figures 8 and 9). This makes our datasets more diverse in terms of explanations compared to existing datasets with house or cycle motifs.
>
> In addition, we have various prediction settings, not just binary classification on whether the motif exist. In the dataset BenLac, we assign labels according to various co-occurrence conditions of either lactam or benzoyl groups. In the dataset BenLacM, we consider the explainability in a multiclass setting in which each class has a separate pattern. This setting is often overlooked by the literature. Our dataset Lactam is a binary classification task, however, in Figure 7, we showed the weight assigned on both the positive and the negative examples, which is also overlooked in the literature.
>
> ```Model performance is sensitive to hyperparameter settings,...```
>
> We generally agree with this notion. Tuning requirement and sensitivity to hyperparameter are reasonably expected in system that is complex (bilevel optimization) with multiple components (explainer and predictor). However, we believe this overhead cost can be amortized over a domain. For example, we use the same setting across multiple molecular datasets and the method obtain generally good results.
>
> ```Just for suggestion, it would be better to have more real-world datasets. For those lacking a ground truth explanation, the fidelity score could be considered.```
>
> We thank the reviewer for the thoughtful suggestion. Fidelity is not relevant for EAGER or ante-hoc explanations in general because we learn both the explainer and the classifier at the same time, not adapting an explainer to an already trained classifier in post-hoc explanation. In ante-hoc explanation, both in graph and other domains, the explainer is learned together with the classifier and is part of the system. As such, producing a prediction requires both the explainer and the classifier. Fidelity requires comparing the differences between predictions using the input graphs and predictions using the explanations. However, in our case, the predictor never predicts based on the input graphs so this metric is not applicable. Instead, we further evaluated the quality of the explanation using a related metric called Reproducibility, which compare how quickly predictive performances drop as we progressively sparsify the explanations. This is reported in Appendix B.
>
> ```Could you elaborate on the rationale for including the average AUC in Table 3?...```
>
> Being an ante-hoc model, EAGER should not only excel in explanation, but also perform well as a predictive model. Averaging AUC across diverse datasets provides a high-level overview of the model's general predictive performance. While it does not capture dataset-specific nuances, it offers a clear benchmark for comparing models holistically
>
> ```Are there plans to include newer baselines in future evaluations? For instance, the addition of MixupExplainer...```
>
> We thank you for referring us to MixupExplainer. So far, in our revision, we have added CAL (2022) and OrphicX (2022). In general, there will always be more baselines that one can add. However, due to the time constraint of the rebuttal period and resource constraints on our side, we are still working on getting more results in. We, still, have added citations and discussion of MixupExplainer in the main text (Line 133-135).

---

> > ### Author Response · Authors · 2024-11-25
> > **Friendly reminder**
> >
> > Dear reviewer,
> >
> > The end of the rebuttal period is coming soon. We would like to hear back from you whether our respond has resolved all of your concerns. If you have any further questions, we are happy to answer.
> >
> > Authors

---

### Official Review · Reviewer_por9 · 2024-11-05

**Soundness:** 2
**Presentation:** 2
**Contribution:** 3
**Rating:** 6
**Confidence:** 4

**Summary:**

Summary:

The paper proposed EAGER - an ante-hoc graph explanation method by optimizing the information bottleneck principle via a bilevel optimization process.

**Strengths:**

Strengths:

1. An ante-hoc graph explanation model is an crucial topic.
2. Introducing a bilevel optimization method is interesting.

**Weaknesses:**

Weaknesses:
1. My primary concern is about the efficiency of the proposed method, particularly given its dual role as both an explanation method and a Graph Neural Network for predicting molecular properties. The efficiency of this method is crucial for its practical application. The authors should thoroughly discuss the computational complexity of their method in the main section of the paper and include experiments on running time. Currently, the assessment of running time is relegated to the appendix and only tested on a relatively small synthetic dataset. This is insufficient to demonstrate the method's efficiency effectively. More comprehensive testing on larger and more diverse datasets is necessary to establish a clearer understanding of the method's performance in real-world scenarios.
2. The effectiveness of the target Graph Neural Network (GNN) model significantly influences the quality of explanations provided. In prior research, particularly with post-hoc explanation methods, it is common practice to maintain a consistent target model across different methods to ensure fair comparisons with baseline approaches. However, due to the unique architecture of the proposed method, it does not use the same GNN classifier as the one employed in the baseline methods. This discrepancy could compromise the fairness of direct comparisons between the proposed method and other baselines, as the underlying GNN model differences might affect the outcome independently of the explanation method's effectiveness.
3. The datasets currently used in the study are relatively small. To more effectively demonstrate the capabilities of the proposed method in classification tasks, it would be beneficial to employ larger datasets, such as HIV or PCBA. Utilizing these more extensive datasets could provide a more robust evaluation of the method's performance.
4. Figure 3 lacks clarity. A more detailed illustration is required to effectively display each component of the process. The figure should aim to distinctly outline and explain the functionalities of each part, ensuring that the figure conveys the intended information clearly and accurately.

**Questions:**

Please refer to weaknesses.

---

> ### Author Response · Authors · 2024-11-20
>
> Thank you for your time and insights in reviewing our paper. We are glad to address your concerns as follows.
>
> ```My primary concern is about the efficiency of the proposed method,... The authors should thoroughly discuss the computational complexity of their method in the main section of the paper and include experiments on running time...```
>
> We thank the reviewer for the thoughtful suggestion. As requested, we have added Section 3.5 that discusses the time complexity of EAGER. Let $C_{GNN}$ be the running cost of the underlying GNN and $C_{hyper}(T,d)$ be the cost of calculating hypergradients, with $T$ being the number of inner iterations and $d$ being the number of dimensions. EAGER's inference time is $O(2C_{GNN})$: an explainer GNN followed by a predictive GNN. During training, EAGER's time complexity is $O((T+1)C_{GNN}+C_{hyper}(T,d))$: one outer iteration followed by T inner iterations plus hypergradient calculation. Notice that when used with a typical GNN, generally $C_{hyper}(T,d)$ scales linearly with batch size, $T$, and $d$ [1]. We report detailed training and inference time in Appendix D.
>
> [1] Amirreza Shaban, Ching-An Cheng, Nathan Hatch, and Byron Boots. Truncated back-propagation
> for bilevel optimization. In The 22nd International Conference on Artificial Intelligence and
> Statistics, pp. 1723–1732. PMLR, 2019.
>
> ```More comprehensive testing on larger and more diverse datasets is necessary to establish a clearer understanding of the method's [running time] in real-world scenarios.```
>
> We generally agree with this suggestion. We would like to point out that since EAGER scales linearly with the number of input graphs, even if we repeat the experiments on larger datasets, we would obtain the same relative comparisons reported in Appendix D. Due to time and resource constraint, we could not repeat the runtime analysis (Appendix D) on larger datasets like HIV or PCBA (due to certain post-hoc baselines taking very long to run). We are doing our best to get more results in. However, we did add HIV to the classification benchmark (see below).
>
> ```... it is common practice to maintain a consistent target model across different methods to ensure fair comparisons with baseline approaches. However, due to the unique architecture of the proposed method, it does not use the same GNN classifier as the one employed in the baseline methods.```
>
> We respectfully disagree and would like to provide more clarification. EAGER is a general framework that can work with any GNN backbone. In our experiments, we tried to the best of our ability to maintain the same GNN architecture across all baselines (see line 419-422). In particular, we rewrote parts of other baselines' source code to produce GNN architectures as similar as possible to the ones we implemented. Moreover, we altered the baselines' source code such that they take in the same atom and bond features as we used. For all the benchmarks and EAGER, if a method requires an underlying GNN backbone, we consistently used GIN. For post-hoc explainers, we also used GIN as the pretrained predictive model.
>
> ```The datasets currently used in the study are relatively small. To more effectively demonstrate the capabilities of the proposed method in classification tasks, it would be beneficial to employ larger datasets, such as HIV or PCBA. Utilizing these more extensive datasets could provide a more robust evaluation of the method's performance.```
>
> We have added HIV as one of the benchmark in Table 3. Additionally, we would like to offer more explanations. The majority molecular datasets are small, especially those curated in real-lab settings. Performances on small datasets with limited data, at the moment, would best reflect expected real-world performances in this domain. That said, we do agree that evaluation on larger datasets is still beneficial in showcasing the method's performance in future uses. Unfortunately, due to time and resource constraint, we could not add more large datasets to the analysis besides HIV.
>
> ```Figure 3 lacks clarity. A more detailed illustration is required to effectively display each component of the process. The figure should aim to distinctly outline and explain the functionalities of each part, ensuring that the figure conveys the intended information clearly and accurately.```
>
> We have updated Figure 3 with more details. Specifically, we put color boxes separating the inner and the outer optimization problems. We also added more descriptive texts on what each component learns, and the training loss used.

---

> > ### Comment · Reviewer_por9 · 2024-11-24
> > **Official response to rebuttals**
> >
> > Thank you to the authors for their effort in addressing my concerns during the rebuttal process.
> >
> > While some of my questions have been partially resolved, my main concern remains regarding the efficiency of the proposed method. The evaluation of training and testing times was conducted on a relatively small binary classification dataset containing only 1,200 graphs. I believe this dataset size is insufficient to fully analyze the method's scalability and efficiency.
> >
> > Although the authors state that EAGER scales linearly with the number of input graphs, it would be important to observe its performance in terms of training, inference, and testing times on a larger dataset. Furthermore, even on this small dataset, EAGER is over 16 times slower than GIN, raising concerns about its practicality for real-world applications.

---

> > > ### Author Response · Authors · 2024-11-24
> > > **Thank you for your response.**
> > >
> > > We would like to thank the reviewer for participating in the discussion. Your feedback is greatly appreciated.
> > >
> > > We would like to clarify that EAGER, as an ante-hoc explainer model, is expected to be more computationally intensive than GIN because we do both classification and explanation at the same time. We would also want to point out that, from Table 7, compared to several other learnable explainers, such as PGExplainer or DIR-GNN, EAGER is significantly faster.
> > >
> > > Regarding the runtime on HIV, we include additional experimental runtime details of EAGER and other models in the following table, which shows similar relative comparisons as in Table 7.
> > >
> > > | Method | Train (s/epoch) | Test (s/fold) |
> > > |--------|:---------------:|:-------------:|
> > > | GCN    | 11.17           | 0.525         |
> > > | GAT    | 11.43           | 0.517         |
> > > | GIN    | 9.26            | 0.502         |
> > > | GSAT   | 13.73           | 0.515         |
> > > | EAGER  | 173.42          | 0.519         |
> > >
> > > From the above table, EAGER is still slower than vanilla GIN (18 times). However, this result is consistent with the reviewer's observation that EAGER is 16 times slower than GIN on Mutagenicity. This confirms that EAGER's runtime linearly grows with the data size, and is bounded by a roughly constant factor relative to GIN's runtime. The inference time is essentially the same across all models.
> > >
> > > We are still working forward gathering data for other explainers and will include these details in the final version. We hope that the updated results so far would help resolve your concern.

---

> > > > ### Comment · Reviewer_por9 · 2024-11-26
> > > > **Please include discussion of efficiency problem in the revised paper, increasing my score**
> > > >
> > > > I sincerely appreciate the additional experiments done by reviewers. Please include the discussion of efficiency problem and running time experiments in the revised paper. I will increase my score to 6.

---

> > > > > ### Author Response · Authors · 2024-11-26
> > > > > **Thank you for your consideration**
> > > > >
> > > > > Dear reviewer,
> > > > >
> > > > > We are glad our answers resolved your concern and greatly appreciate your re-adjustment to the score. We will further update the manuscript once we gathered all the numbers.

---

### Official Review · Reviewer_QTBb · 2024-11-07

**Soundness:** 2
**Presentation:** 2
**Contribution:** 2
**Rating:** 5
**Confidence:** 2

**Summary:**

The authors propose to learn an edge weighting scheme together with a graph neural network where the edge weights serve as explanation of the graph neural network. The combined training of the explainer and the GNN minimizes an Information Bottleneck objective to reduce the size of the explanations while maximizing the predictive performance of the GNN learner. Empirical experiments on suitably preprocessed datasets suggest that the method, called EAGER, works well in practice.

**Strengths:**

- The paper is well structured and introduces all relevant concepts and steps
- Ante-hoc explainers -- in this case subgraphs on which the GNN model is allowed to learn solve several of the problems of instance based post-hoc explanations of graphs
- the overall architecture seems simple and elegant

**Weaknesses:**

- It remains unclear from the presentation whether indeed subgraphs are used or whether the explainer computes an edge weight that just scales down edge attributes during training and/or inference. In the latter case, explanations would be not much helpful, I fear.
- The edge weighting approach to arrive at a subgraph(?) is not expressive enough to capture many phenomena that are taking place in graphs. See question below.
- It remains unclear how to control the size of the explanations/subgraphs

**Questions:**

# Statement
I am terribly sorry about my lapse. There is really no excuse for posting the wrong review here and then not reacting to multiple questions here. I have changed it now, but please, ignore my questions and comments, as there is really no time left to act on them. I accept full responsibility and am truly sorry.

# Questions

- Can you please be more precise about the usage of the edge weights in training and inference? Is $\alpha$ in Algorithm 1 a hard threshold that removes all edges with weight $<\alpha$? How to choose this?
- Assuming thresholding takes place: Is precision at 10 or ROC a good evaluation measure? In this case, I assume that one has no influence on the amount of edges that is selected by the explainer.
- Assuming no thresholding takes place: How can you ensure that the GNN after edge weighting only uses information of high weight edges, as claimed in the introduction. In this case, it seems that message passing uses all existing edges of the graph and may also reweight low weight edges from the explainer with suitable parameters.
- Furthermore both p@10 and ROC at some point require to select a threshold. Does this imply that the user needs to know/set the size of the explanations that they want to get?
- The explainer model seems to weight edges independently of graph topology, just based on attributes of the edge and the two incident nodes. This, however, implies that such an explainer cannot distinguish e.g. a C-C edge on a six-cycle from a C-C edge on a three-cycle. Hoever, it seems, that this is the case in Figure 1c. Are you using some particular preprocessing to add this information?

# Minor issues and typos
- l75 We introduces
- l203 two distributions are keps
- Algorithm 1 / Section 3.4.2 use inconsistent notation. While in Alg.1 $\alpha$ appears as threshold parameter, it appears as a tradeoff parameter in a different place. I suggest to rename one of the alphas and to consistently use the same sybmol for the threshold parameter in Alg.1 and Sec.3.4.2

---

> ### Author Response · Authors · 2024-11-20
> **Mistaken Review?**
>
> Dear reviewer,
>
> We believe you have mistakenly submitted a review for another paper. Should you resubmit the correct review, we are happy to address any of your concerns and recommendations. Looking forward to hearing from you.

---

### Comment · Area_Chair_xSVN · 2024-11-22

Hi reviewers,

The authors have posted their rebuttals. Could you please check their responses and engage in the discussions? Please also indicate if/how their responses change your opinions.

Thanks,

AC

---

### Meta-Review · Area_Chair_xSVN · 2024-12-21

**Metareview:**

This paper proposes an ante-hoc graph explanation method, EAGER, by optimizing the information bottleneck principle within a bilevel optimization framework. EAGER assigns influence values to edges, which are used to modify the input graph by scaling the adjacency matrix and creating an influence-weighted GNN. Thus, the explanations and the model are jointly learned together. Ante-hoc explanation is important and most existing works are for post-hoc explanation. Thus, this work is well-motivated and tackles some limitations of existing studies. However, even though two reviewers champion the paper, there remain significant concerns, including the use of simple datasets and easy tasks for evaluation, the lack of fidelity as a metric which is important, the lack of novelty compared with existing works especially existing ante-hoc methods, presentation issues, etc. I believe this paper has value to the community; however, I encourage the authors to carefully check the comments and significantly revise the paper for a future conference.

**Additional Comments On Reviewer Discussion:**

Reviewer QTBb posted a review for a different paper. I sent several reminders to request the right review but there was no response, so Reviewer QTBb is not considered for decision. Reviewer por9 seemed satisfied with the authors' response. The other three reviewers share similar concerns, some of which are not acknowledged by or convincing to the reviewers. Hence, I hope the authors can make great efforts to tackle the commonly shared concerns for future submission.

---

### Decision · Program_Chairs · 2025-01-22

Reject